# Blindfolded Experts Generalize Better: Insights from Robotic Manipulation and Videogames

**Ev Zisselman,\* Mirco Mutti, Shelly Francis-Meretzki, Elisei Shafer, Aviv Tamar**
Technion – Israel Institute of Technology

## Abstract

Behavioral cloning is a simple yet effective technique for learning sequential decision-making from demonstrations. Recently, it has gained prominence as the core of foundation models for the physical world, where achieving generalization requires countless demonstrations of a multitude of tasks. Typically, a human expert with full information on the task demonstrates a (nearly) optimal behavior. In this paper, we propose to hide some of the task's information from the demonstrator. This "blindfolded" expert is compelled to employ non-trivial *exploration* to solve the task. We show that cloning the blindfolded expert generalizes better to unseen tasks than its fully-informed counterpart. We conduct experiments of real-world robot peg insertion tasks with (limited) human demonstrations, alongside videogames from the Procgen benchmark. Additionally, we support our findings with theoretical analysis, which confirms that the generalization error scales with $\sqrt{I/m}$, where $I$ measures the amount of task information available to the demonstrator, and $m$ is the number of demonstrated tasks. Both theory and practice indicate that cloning blindfolded experts generalizes better with fewer demonstrated tasks. Project page with videos and code: https://sites.google.com/view/blindfoldedexperts/home.

## 1 Introduction

Behavioral cloning (BC) is a simple yet effective method for training policies in sequential decision-making problems [32, 6]. In BC, an expert demonstrates how to perform a task, and the sequence of observation-action data is input to a supervised learning algorithm for training a policy.

A key question in BC is generalization—how many demonstrations are required to train an effective policy. For single tasks, a well-investigated challenge is compounding errors—small mistakes in the trained policy may lead to visit states that the expert did not visit, further increasing the prediction errors [43]. However, recent results show that using appropriate neural-network architectures, BC can learn to solve complex tasks even with a modest number of demonstrations [6, 59], and these results are reinforced by recent theory [13]. For multiple tasks (or significant variations of a single task), on the other hand, BC still requires abundant data, and recent methods for mitigating data requirements include augmentations [26], simulation [52], and fine tuning foundation models trained on large scale demonstration data [30, 49, 21]. In this work, we hence focus on generalization to task variations.

While various works study how to improve generalization via the BC algorithm [43], the policy representation [6], and the data diversity [23], one aspect that remains unexplored is *the experts themselves*. Many tasks can be solved in various ways—can some behaviors generalize better than others? Recently, in the context of zero-shot reinforcement learning, Zisselman et al. [61] showed that certain exploratory behaviors generalize better than goal-oriented, reward-maximizing behavior. Intuitively, since exploratory behavior is less goal-oriented, it is less dependent on any particular task

---

\*Correspondence E-mail: ev_zis@campus.technion.ac.il

39th Conference on Neural Information Processing Systems (NeurIPS 2025).

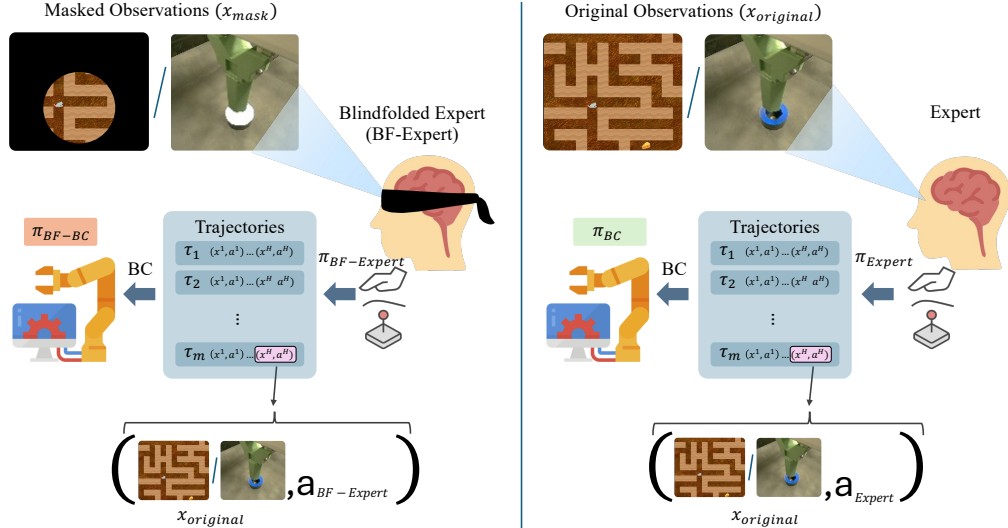

Figure 1: Illustration of the learning process. Note that the mask on observations only applies to the blindfolded expert, while the observations in the logged trajectories are unmasked in both cases.

instance and therefore, more likely to generalize to novel tasks. In this work we ask—can a similar principle be useful also for imitation learning?

Our main idea (depicted in Figure 1) is that by introducing a *blindfold*—an information bottleneck on the expert's observation that makes it harder to identify the particular task, we can induce the expert to express a more exploratory and less task-dependent behavior, which we conjecture will generalize better. Importantly, *our method does not change the observations used for training the policy* but only the expert's behavior, and is therefore compliant with any BC algorithm, and complementary to the methods for improving generalization mentioned above. Since the expert's exploratory behavior is typically history dependent, our method makes use of policy architectures that can process a sequence of observations, such as recurrent neural networks or transformers [7, 53].

Theoretically, we prove an upper bound on the generalization that scales with $\mathcal{E}_{gen} + \sqrt{I/m}$, where $I$ measures the amount of task information available to the demonstrator, $m$ is the number of demonstrated tasks, and $\mathcal{E}_{gen}$ is a cost associated with the expert not taking optimal actions. For domains where the exploratory behavior can still solve the task, $\mathcal{E}_{gen}$ is zero, and thus by lowering $I$ using a "blindfold" we reduce the generalization error without paying any price. To our knowledge, this result is the first of its kind in relating non-trivial properties of the expert's behavior to multi-task generalization of the resulting BC policy.

Empirically, we demonstrate our approach on simulated games from the Procgen suite [8]—a standard benchmark for task generalization, and on a real-robot peg insertion task, based on the FMB challenge [25], where the shapes of the peg and the hole define the task. For the Procgen games, our blindfold hides the observation and reveals only the agent's immediate surroundings to the expert. For the peg insertion domain, we let the expert teleoperate the robot by observing images from robot-mounted cameras, and mask out the hole shape from the image. In both domains, we find that the blindfold induces more exploratory behavior from the expert, which in turn yields significantly better generalization to different tasks.

Our results pave the way to a new and principled approach for collecting demonstrations, both for specific problems, and also for more general foundation-model scale endeavors.

## 2   Problem formulation

Throughout the paper, we will focus on a *multi-task* imitation learning setup in which we aim to clone an expert's behavior from demonstrations of a (small) selection of tasks with the goal to generalize the behavior to (many) unseen tasks. As we shall see both theoretically and empirically, the generalization is not only affected by the number of available demonstrations, but also by the information available to the expert when performing demonstrations. First, let us introduce the setting formally.

**Setting.** We consider a set of tasks $\Theta := \{\theta_i\}_{i=1}^M$ and a task distribution $P_0 \in \Delta(\Theta)$, where $\Delta(S)$ denotes the probability simplex over a set $S$. Each task $\theta \in \Theta$ is defined through a Markov Decision Process (MDP [33]) $\mathcal{M}_\theta := (\mathcal{X}, \mathcal{A}, p_\theta, r_\theta, H)$, where $\mathcal{X}, \mathcal{A}, H$ respectively denote the observation space, the action space, and the horizon of an episode, which we assume common across all the tasks in $\Theta$.[2] Instead, each task may have their own transition model $p_\theta : \mathcal{X} \times \mathcal{A} \to \Delta(\mathcal{X})$ and reward function $r_\theta : \mathcal{X} \times \mathcal{A} \to [0, 1]$. A history-based randomized policy is a sequence of functions $\pi := \{\pi_h : \mathcal{T}_h \to \Delta(\mathcal{A})\}_{h=0}^{H-1}$ where $\mathcal{T}_h$ is the set of $h$-steps trajectories $\tau^h = (x^0, a^0, r^0, \ldots x^h)$ and $\mathcal{T} = \cup_{h=0}^{H-1} \mathcal{T}_h$. A policy $\pi$ on the MDP $\mathcal{M}_\theta$ induces a distribution $\mathbb{P}_\theta^\pi$ over trajectories with the following process. An initial observation is sampled $x^0 \sim p_\theta(\cdot)$. Then, for every step $h \geq 0$, an action is sampled from the policy $a^h \sim \pi(\tau^h)$, the reward $r^h = r_\theta(x^h, a^h)$ is collected, and the MDP emits the next observation $x^{h+1} \sim p_\theta(x^h, a^h)$. The process goes on until the step $H$ is reached.[3] The Reinforcement Learning (RL [47]) objective for an MDP $\mathcal{M}_\theta$ is the cumulative sum of rewards $J(\pi) := \mathbb{E}_{\mathbb{P}_\theta^\pi}[\sum_{h=0}^{H-1} r^h]$, where the sequence $(r^0, \ldots r^{H-1})$ is taken on expectation over trajectories $\tau \sim \mathbb{P}_\theta^\pi$. An optimal policy for $\mathcal{M}_\theta$ is denoted as $\pi^* \in \arg\max J(\pi)$. For some $R \in \mathbb{N}$, we assume $J(\pi^*) \leq R$, where typically $R = 1$ when rewards are sparse, as large as $H$ when rewards are dense.

**Behavioral cloning.** In the setting described above, we assume to have access to a dataset of expert demonstrations $E = \{\theta_i \sim P_0, (\tau_{i1}, \ldots \tau_{in}) \sim \mathbb{P}_{\theta_i}^{\pi^E}\}_{i=1}^m$ where $\tau_{ij} = (x_{ij}^0, a_{ij}^0, r_{ij}^0 \ldots x_{ij}^H, a_{ij}^H, r_{ij}^H)$ is a $H$-steps trajectory sampled independently from a policy $\pi^E$ in the MDP $\theta_i$. Thus, the total number of trajectories is $|E| = mn$ and the total number of transitions is $mnH$. With the available data, we aim to clone the expert's behavior $\pi^E$, a problem that is known as *behavioral cloning* [41]. The idea is to train a policy $\widehat{\pi}$ to mimic the expert's policy $\pi^E$ by minimizing a supervised learning loss on the demonstrations. While several choice of loss functions could be made [55], here we opt for the negative log likelihood as in [13]. The behavioral cloning problem is then

$$\widehat{\pi} \in \arg\min_{\pi \in \Pi} \ \mathcal{L}(\pi) := \sum_{i=1}^m \sum_{j=1}^n \sum_{h=0}^{H-1} \log\left(\frac{1}{\pi(a_{ij}^h | \tau_{ij}^h)}\right) \tag{1}$$

where $\Pi$ is a policy space of our choice and $\tau_{ij}^h$ is $h$-steps chunk of the trajectories $\tau_{ij}$ in the dataset of demonstrations $E$, $a_{ij}^h$ is the action taken at step $h$ in $\tau_{ij}$. While a sufficiently expressive policy space $\Pi$ may allow for a cloned policy $\widehat{\pi}$ that closely approximates the expert on the training data $\mathcal{L}(\widehat{\pi}) \approx 0$, we typically aim for a policy $\widehat{\pi}$ that can mimic the expert's behavior on unseen data as well. Differently from the common setting [41, 42, 56, 38, 36, 37, 13], here we are not only concerned with *generalization* across unseen observations in $\mathcal{X}$, but also across unseen tasks in $\Theta$. Before proceeding with the study of generalization in the next section, we introduce additional notation for later use.

**Additional notation.** In our behavioral cloning problem (1), a single data point is given by the triplet $(\theta_i, \tau_{ij}^h, a_{ij}^h)$, which we intend as realizations from the random variables $(T, X, A)$ distributed as $T \sim P_0$ and $(X, A) \sim \mathbb{P}_T^{\pi^E}$ respectively. We will turn to one or the other notation when convenient. For a random variable $A$ taking values $a_1, a_2, \ldots$ with probabilities $p(a_1), p(a_2), \ldots$, we denote its *entropy* $H(A) = -\sum_i p(a_i) \log p(a_i)$. For two random variables $A, B$, we denote their *mutual information* $I_{A;B} = H(A) - H(A|B) = H(B) - H(B|A)$, where $H(A|B)$ is the conditional entropy. Finally, we will use the symbol $\lesssim$ to hide constant and lower order terms from inequalities.

## 3 Generalization analysis

In the previous section, we detailed how an expert's policy can be "cloned" from data by solving the optimization problem (1). Obviously, fully cloning the expert's behavior is a far fetched objective when limited demonstrations are available: When training data spans only a small portion of the observation space $\mathcal{X}$ and the set of tasks $\Theta$, how can we extract information on what would the expert do in unseen observations and tasks? Nonetheless, we aim for our cloned policy $\widehat{\pi}$ to *transfer* at least part of the expert's behavior beyond the demonstrated observations and tasks. In this section, we

---

[2] Note that this does not hinder generality, as we can always take $\mathcal{X} = \cup_{\theta \in \Theta} \mathcal{X}_\theta$ when observation spaces vary across tasks (ditto for the action space) and $H = \max_{\theta \in \Theta} H_\theta$ when the episode horizons vary.

[3] Oftentimes, the episode horizon is an upper bound to the episode length, while secondary termination conditions may end the episode early, as it will be the case in our experimental setting. For the ease of presentation, we ignore early termination in our setup and consider episodes of length $H$.

provide a formal study of the *generalization* guarantees of the cloned policy $\widehat{\pi}$, showing an original dependence with the *information* available to the expert when collecting demonstrations. To specify what do we mean by "information" in this setting, let us consider the figure below.

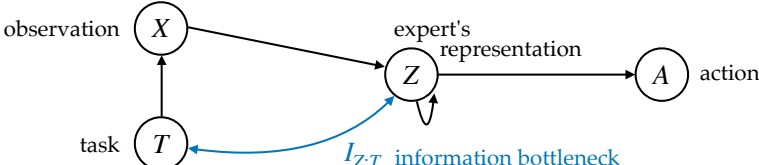

The latter is a graphical illustration of the expert's behavior.[4] At each step, the expert takes as input an observation ($X$) which depends on the task they are going to demonstrate ($T$), processing them into an internal representation $Z$. The demonstrated action ($A$) is then conditioned on $Z$. Note that $Z$ is recurrent and retains any information that is relevant to select $A$, including the task information that may be available in the observation $X$. The mutual information $I_{Z;T}$ measures the task-related information that goes into $Z$ and, consequently, how much the strategy to select $A$ relies on it.

We typically expect an expert with full information on the task—large $I_{Z;T}$—to demonstrate an optimal policy specific to the task, without any exploratory actions. Instead, whenever there is an information bottleneck between the task information and the expert—small $I_{Z;T}$—we expect them to take exploratory actions to first "understand" the task in order to solve it. We still call them *experts* as we assume their behavior to be optimal *with the given information*, a concept that has been formalized with Bayes-optimal policies [16]. We conjecture that the latter behavior may generalize better to new tasks, as the process of understanding the task is more general than just solving it. In the following, we provide a formal result based on this conjecture, in which we analyze the generalization gap of the cloned policy as a function of the information $I_{Z;T}$ available to the demonstrator.

Analogously to previous works [e.g., 41, 13], we are interested in deriving an upper bound on the performance gap between an optimal policy for each task ($\pi^*$) and the cloned policy ($\widehat{\pi}$). Since we are considering a multi-task setting, we average the gap across the task distribution $P_0$,[5]

$$\underset{T\sim P_0}{\mathbb{E}}[J(\pi_T^*) - J(\widehat{\pi})] = \underset{T\sim P_0}{\mathbb{E}}\left[\underset{\mathbb{P}_T^{\pi^*}}{\mathbb{E}}\left[\sum_{h=0}^{H-1} r^h\right] - \underset{\mathbb{P}_T^{\widehat{\pi}}}{\mathbb{E}}\left[\sum_{h=0}^{H-1} r^h\right]\right]. \tag{2}$$

Before stating the result, we introduce a few technical assumptions. First, we define the *generalization error* of a policy $\pi$ as

$$\mathcal{E}_{gen}(\pi) := \underset{T\sim P_0}{\mathbb{E}}\,\underset{XA\sim\mathbb{P}_T^{\pi^*}}{\mathbb{E}}[\mathbf{1}(\pi(X) \neq A)] \tag{3}$$

for a single point indicator loss. We make the following assumptions on the expert's policy.

**Assumption 1.** *The expert's policy $\pi^E$ is deterministic.*

**Assumption 2.** *The generalization error of the expert's policy is given by $\mathcal{E}_{gen}(\pi^E)$.*

Note that we allow the expert's policy to depend on the history, for which assuming determinism is reasonable even when an information bottleneck is applied to the expert. Whereas it is standard in the literature to assume the expert is optimal, i.e., $\mathcal{E}_{gen}(\pi^E) = 0$, in the presence of an information bottleneck we do not take for granted that the expert is optimal *in all the tasks*. Nonetheless, if the expert's behavior is Bayes-optimal, non-trivial worst-case bounds on $\mathcal{E}_{gen}(\pi^E)$ hold [5], for which the generalization error only scales with $\log(H)$ under our assumptions. Moreover, in settings where the reward is sparse denoting task success, i.e., $R = 1$, the Bayes-optimal policy may still have $\mathcal{E}_{gen}(\pi^E) = 0$ w.r.t. some optimal policy, albeit inefficient in the number of steps.

Then, regarding the behavioral cloning problem (1), we make a pair of assumptions as follows.

**Assumption 3.** *The expert's policy is* realizable *in the policy space $\Pi$, i.e., $\pi^E \in \Pi$.*

**Assumption 4.** *We have access to an optimization oracle that solves problem* (1) *with bounded error*

$$\mathcal{E}_{opt}(\widehat{\pi}) := \frac{1}{mnH}\sum_{i=1}^{m}\sum_{j=1}^{n}\sum_{h=0}^{H-1}\mathbf{1}(\widehat{\pi}(\tau_{ij}^h) \neq a_{ij}^h).$$

---

[4]Note that this is not related to the architecture of the cloned policy, which will be discussed later on.
[5]Note that the optimal policy $\pi^*$ depends on the task $T$ whereas $\widehat{\pi}$ is a single policy cloned from data.

In principle, one can fulfill Asm. 3 by cloning the demonstrations into a rich enough policy space $\Pi$. However, a more expressive policy space may lead to a harder optimization problem, especially when the policies are represented through large neural networks, for which (1) is non-convex.

We now have all the ingredients to state our main result.

**Theorem 3.1.** *For a confidence $\delta \in (0, 1)$, it holds with probability at least $1 - 2\delta$*

$$\mathbb{E}_{T \sim P_0}[J(\pi_T^*) - J(\widehat{\pi})] \lesssim RH \left( \mathcal{E}_{gen}(\pi^E) + \mathcal{E}_{opt}(\widehat{\pi}) + \sqrt{\frac{I_{T;Z}|\mathcal{A}|\log(|\mathcal{A}|/\delta)}{m}} + \frac{\log(|\Pi|m/\delta)}{n} \right).$$

All of the derivations and the hidden constants can be found in Appendix A. Instead, here we unpack the bound and discuss the meaning of each term. First, the $RH$ factor accounts for the cost of a "mistake" of the cloned policy, i.e., choosing an action different from $\pi^*$. The terms $\mathcal{E}_{gen}(\pi^E)$ and $\mathcal{E}_{opt}(\widehat{\pi})$ depends on the quality of the expert's policy and the solver for (1), hence they cannot be reduced with additional data. The last term, scaling with the number of trajectories in each demonstrated task $n^{-1}$, comes from a typical behavioral cloning analysis of generalization within the training task [e.g., 13]. The more demonstrations we have from a task, the better we can clone the expert's policy in that task. The third term, scaling with the number of demonstrated tasks $m^{-1}$, controls the generalization across tasks and comes from the analysis of generalization induced by an information bottleneck [20], which is expressed by $I_{Z;T}$. The most important finding of our result lies in this term: To improve generalization of the cloned policy, we can either increase the number of tasks $m$ or apply an information bottleneck—a "blindfold"—to the demonstrator to reduce $I_{Z;T}$ without paying any meaningful price, as it is typically $\mathcal{E}_{gen}(\pi^E) = 0$ for tasks with sparse rewards and $\mathcal{E}_{gen}(\pi^E) \lesssim \log(H)$ for dense rewards, while other terms remain the same.

In the next sections, we provide an extensive empirical evaluation showing that this result is far from being a theoretical fluke, but translates to practical scenarios as well.

**Overcoming assumptions.** The result presented in this section holds for deterministic expert's policies (Asm. 1), finite action space (due to $|\mathcal{A}|$ dependency), and finite policy class (due to $|\Pi|$ dependency). Deterministic expert's policy and finite policy class can be easily overcome by extending the in-task generalization result to stochastic policies and infinite policy classes, as done in [13]. Instead, the dependency on $|\mathcal{A}|$ comes from reducing the cloning problem to classification, in order to invoke information bottleneck generalization results [20]. In principle, extending the analysis to continuous action requires an analogous generalization bound for the regression problem [29] and to circumvent established negative results for this setting [46].

## 4 Experiments

In this section, we report an experimental campaign to validate the results of previous sections and to demonstrate the importance of the expert's behavior for multi-task BC generalization. To this end, we train two policies, with the same BC algorithm, on human demonstrations collected by either a traditional expert or a blindfolded expert[6]. We refer to the resulting policies as $\pi_{BC}$ and $\pi_{BC-BF}$ respectively. We compare them in the success rate achieved on both the demonstrated tasks and unseen test tasks. We repeat the experiment twice, first in simulation on the Procgen maze and heist (Section 4.1), then on a real robot peg insertion task (Section 4.2). For both domains, we describe how the information bottleneck for the blindfolded expert is obtained in practice.

### 4.1 Procgen maze and heist

Procgen [8] is a popular benchmark for measuring sample efficiency and generalization [35, 34, 9, 22, 61]. It consists of 16 different procedurally generated environments in which new levels are randomly generated for every episode, forcing agents to handle changing layouts, colors, and textures. Here we focus on the maze and heist games in the "easy" setting, in which each level is a 2D maze-like layout that the agent navigates. These tasks are the most challenging tasks for generalization in the Procgen suite, and until the results of [61] have seen only minor improvements over random walk. We test how training on a set of demonstrated maze and heist tasks generalizes to unseen tasks for the different experts.

---

[6]Our human demonstration dataset for Procgen maze and heist is available on the project website.

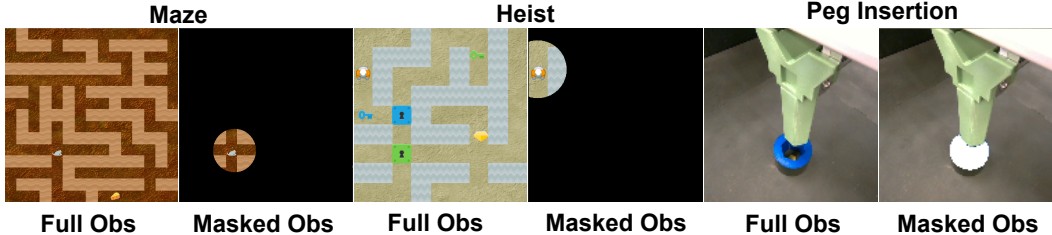

Figure 2: Demonstration of actual experts' observations. For the Procgen maze (left-pair), Procgen heist (middle-pair), and robotic peg insertion (right-pair). We show the full observation of the Expert and the masked observation of the Blindfolded-Expert.

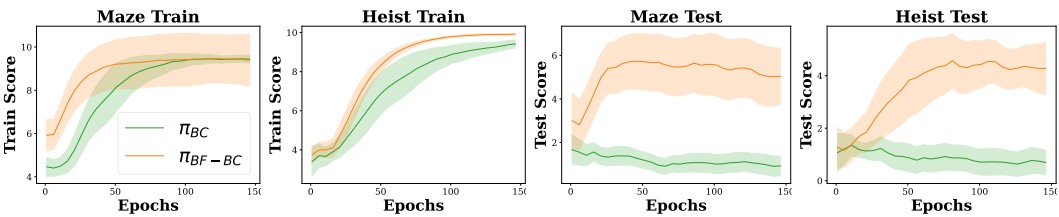

Figure 3: Game score as a function of training epochs in the Procgen maze and heist. Left-pair: performance on 200 training levels. Right-pair: performance on unseen test levels. The mean and standard deviation are computed over 10 seeds.

**Maze.** In Procgen maze, the agent (mouse) must find and reach a single goal (piece of cheese) in a 2D maze. Procedural level generation produces variation in maze layout (sprawling corridors), goal positions, and the color and texture of navigable spaces.

**Heist.** The Procgen heist is based on a maze-like level, in which the agent has to additionally solve a sequence of sub-goals in order to solve the task (reaching the gem). At each level, the goal (gem) is hidden behind a series of color-coded locks, accessible through keys of corresponding colors, that are scattered throughout the level (see Figure 2). This task requires multi-stage planning in the form of lock-waypoints and to overcome variations in sub-task ordering, making the task more complex and challenging than a regular maze.

**Setup.** For training, we consider 200 procedurally generated levels. There are 4 discrete **actions**, move up, down, left, and right in the maze game, and additional 4 actions in heist: left-down, left-up, right-down, and right-up. The **observations** are $64 \times 64$ RGB images of the current state as a top-down view of the maze.

**Data collection.** We collect a total of $4K$ demonstrated trajectories from 200 different training levels (20 trajectories per level). The environment allows for at most 500 input steps for maze and 1000 for heist, and we only retain trajectories of successful expert demonstrations. The **standard experts** are humans who see the top-view of the entire game, and therefore are likely to follow the shortest path to the goal. The **blindfolded experts** are humans playing the game with occluded observations of its layout, such that only the immediate proximity of the agent is visible and the rest of the level is concealed (see Figure 2, middle and left). As a result, human experts cannot directly plan a path to the goal location, and exploration of the level is needed. Note that while the observations are masked to the expert, the stored data contains the original (unmasked) observations for training the cloning algorithm. The **trajectories** are tuples of observation, action, reward, and done flag $(o_t, a_t, r_t, \text{done})$.

**Policy architecture and training.** Training is conducted from scratch on the demonstrated trajectories by minimizing the negative log likelihood 1. We use the architecture from [27] for both the Expert ($\pi_{BC}$) and BF-Expert ($\pi_{BF-BC}$)—a ResNet [17] to encode the observations, which are then processed by two fully-connected layers. To capture the exploratory behavior, we add a single GRU [7] before the Softmax policy layer (further details are in Appendix C).

Table 1: Average number of steps $\pm$ std in the trajectories demonstrated by standard Experts and BF-Experts. The latter takes more steps due to the information bottleneck, which forces exploration.

| Mode | ⬢ | ★ | ■ | ✚ | ◀ | Maze |
|---|---|---|---|---|---|---|
| Experts | $61.2 \pm 1.4$ | $69.4 \pm 3.3$ | $64.8 \pm 2.1$ | $96.6 \pm 4.9$ | $68.8 \pm 2.2$ | $28.6 \pm 1.2$ |
| BF-Experts | $74.3 \pm 5.9$ | $165.6 \pm 42.9$ | $163.6 \pm 43.6$ | $148.6 \pm 33.1$ | $194.4 \pm 55.0$ | $51.6 \pm 1.8$ |

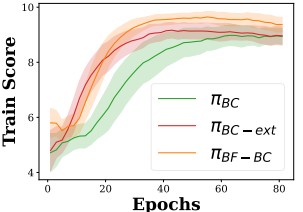 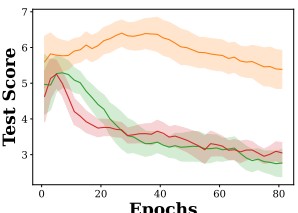

Figure 4: Procgen maze score as a function of training epochs. Left: performance on 100 training levels. Right: performance on unseen test levels. Policies $\pi_{BC}$ and $\pi_{BF-BC}$ are trained on $2K$ trajectories, the former clearly overfits. We also train $\pi_{BC-ext}$ on $4.6K$ non-blindfolded experts' trajectories, matching the total number of steps of $\pi_{BF-BC}$. The resulting $\pi_{BC-ext}$ overfits despite access to twice the amount of data. The mean and standard deviation are computed over 10 seeds.

**Results.** In Figure 3, we compare the game score over the training epochs achieved by policy cloning of the standard Expert ($\pi_{BC}$) and the BF-Expert ($\pi_{BC-BF}$). As we can see from Figure 3 (left-pair), the performance of the two policies is similar on the training levels, approaching the maximum task score. However, as evident from Figure 3 (right-pair), while the $\pi_{BC-BF}$ gracefully improves the test score, the $\pi_{BC}$ policy overfits to the training-set, and its test score slightly degrades with further training. This is a testament to the inherent generalization capabilities granted by the BF-Expert w.r.t. a standard Expert.

**Number of steps.** Table 1 (left-most column) details the average number of steps taken by the experts (Experts) and the blindfolded experts (BF-Experts) across 100 demonstrated levels, accruing $2K$ maze trajectories. Trajectories taken by the Experts are shorter on average than the BF-Experts, supporting the assumption that the BF-Experts exhibit a more exploratory behavior, whereas the Experts take the shortest path to the goal.

To further highlight the contribution of exploration toward generalization and to show that it isn't merely the result of additional training steps, we train a policy $\pi_{BC-ext}$ on double the amount of non-blindfolded experts' trajectories—to match the total number of steps produced by the blindfolded experts. Figure 4 shows that even with an equal number of total steps from both experts, $\pi_{BC-ext}$ overfits. Further details on the number of trajectories and steps are provided in Appendix C.

## 4.2 Robotic peg insertion

Peg insertion is a standard problem in robotic manipulation. Here, we consider the insertion task in the Functional Manipulation Benchmark [FMB, 25], which focuses on inserting variously shaped pegs into tightly matching holes. Different from [25], however, we investigate *generalization*: How training on a fixed set of shapes generalizes to inserting previously unseen shapes. We simplify less relevant technical aspects of the benchmark by fixing the peg to the robot gripper, and 3D-printing individual holes, so that discerning the target hole becomes trivial (see Figure 5-left). In addition, we added several new shapes to the benchmark to increase its variations.

**Setup.** The task comprises ten pegs of various shapes with corresponding slots. In our setting, the robot initiates with the peg already in its grip and needs to insert it into a single-slotted board anchored to the surface. The **robot setup** is shown in Figure 5 in full, alongside a few examples of peg shapes. We use a Franka Emika Panda robot arm and teleoperate the robot using a SpaceMouse. For operating the robot and the SpaceMouse, we use the SERL open-sourced package [24] with the same settings as the authors. The **actions** are input as SpaceMouse commands: 6-DoF end-effector move and twist (location and Euler angles) at 10Hz, tracked by a low-level impedance controller running at 1KHz. The **observations** are obtained as RGB-only images from two Intel RealSense

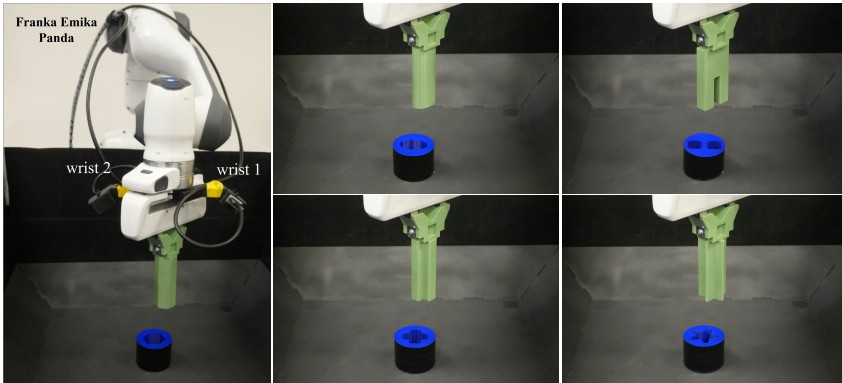

Figure 5: The robotic arm configured for peg insertion (left). Close-up view of various peg shapes and their boards (right).

D405 cameras, mounted on the robot end-effector, which simultaneously capture images[7]. To avoid background distractions in image observations, we place each shaped board inside a black bin. In addition to the 6-DoF pose of the robot end-effector, the framework also includes the force, torque, and velocity information provided by the Franka Panda robot.

**Data collection.** We split the ten shapes into training and test shapes. When collecting demonstrations, only the training shapes are used, while the test shapes serve as a withheld subset for evaluation. We collect 400 trajectories from human demonstrations of each training shape, first with a standard Experts and then with BF-Experts. Each trajectory begins at a random initial pose. When the full information from the wrist cameras is provided to the human experts, they easily succeed in inserting all training shapes. The **blindfolded experts** are human experts exposed to redacted observations from the wrist cameras, which occlude the articulation of the slot as they attempt to insert the peg (see Figure 2). We use SegmentAnything2 (SAM2) [39] modified to segment a live video stream and prompt it to mask out the shape of the target hole. In addition to the masked-out images from the wrist cameras, the initial robot pose varies with each insertion attempt, thus preventing the expert from memorizing or inferring the articulation of the target hole. Note, however, that the recorded trajectories collected by both the Expert and BF-Expert contain the full unmasked observation. The distinction is then the behavior of the two experts, with blindfolded experts taking exploratory actions to cope with masked-out images in an attempt to complete the task. The **trajectories** are collected on the training shapes only. Table 1 details the training shapes and the average number of steps taken by the standard Expert and BF-Expert until successful peg insertion. Clearly, the trajectories demonstrated by the BF-Expert are longer, indicating more exploratory behavior, whereby the expert must rely on masked-out images until resolving the correct articulation for inserting the peg. In the next sections, we show that the resulting exploratory behavior is useful for generalization.

**Policy architecture and training.** Training was conducted for $k \in \{2, 3, 4, 5\}$ peg shapes, with the remaining shapes serving as a withheld test set. We use the same network architecture for cloning both the expert $\pi_{BC}$ and the blindfolded expert $\pi_{BF-BC}$. Specifically, we use a weight-shared frozen ResNet-10 encoder [17] pretrained on the ImageNet dataset [10] for encoding the incoming images from both wrist cameras. The resulting embeddings are concatenated with the MLP-embedding of the proprioceptive information before entering a single GRU [7] that outputs a Gaussian policy. The use of a memory-based architecture is crucial to fully capture the non-Markovian exploratory behavior of the blindfolded expert [28]. For more detailed specifications regarding our experimental setup and hyperparameters, please refer to Appendix B.

**Results.** Figure 6 shows the success rate for $k = 2, 5$ training shapes (results for $k = 3, 4$ are in Appendix B). The success rate is the average of 24 insertion attempts per peg shape by the robotic arm. The results demonstrate that cloning the BF-Expert with $\pi_{BF-BC}$ achieves better generalization compared with the standard Expert, cloned with $\pi_{BC}$, across all peg shapes and over all test subsets. Importantly, the advantage of the proposed blindfolding approach is more significant when fewer

---

[7]Following the conclusions of FMB [25], we omit the depth data, as they showed it has a marginal benefit.

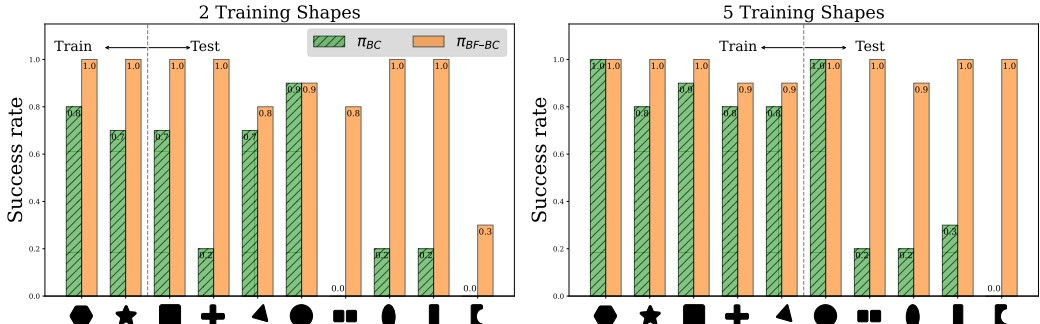

Figure 6: Success rate of robotic peg insertion for 10 peg shapes (horizontal axis). We train on a subset of shapes $k = \{2, 5\}$, with the remaining shapes withheld as test-set. Both for $k = 2$ (left) and $k = 5$ (right), cloning blindfolded experts generalizes better than cloning standard experts.

shapes are used to train the model, i.e., when a larger portion of shapes are withheld during training. Interestingly, the figure shows that even for shapes encountered in training, the $\pi_{BF-BC}$ policy generalizes better than cloning the standard expert $\pi_{BC}$. This is a result of the limited ability of expert demonstrations to account for all real-world variations, e.g., lighting and control-loop errors. In comparison, the $\pi_{BF-BC}$ is more robust to these kinds of errors, since it imitates blindfolded experts who cannot rely on visual cues, and must compensate in order to solve the task. Additionally, it is worth mentioning that certain shapes are more challenging than others. We observe that shapes with greater radial symmetry and convexity (such as the circle, triangle, hexagon) are simpler to learn than shapes with non-radial symmetry (such as the ellipse, rectangle) or non-convex shapes (such as the cut-out-rectangle), as they require a more specific orientation for insertion.

**The sensitivity to the choice of a specific blindfold.** To demonstrate the generality of our approach, we conduct an experiment of robotic peg insertion, where we apply random noise to the experts' observations instead of masking the shape socket. We collected a total of 1600 expert trajectories with random noise $p \sim U[0, P]$ added to all pixel values, from varying maximum noise level $P \in [0, 100, 170, 200]$ and 2 training shapes (square and star). We then train a network on each noise level $P$ (i.e., 400 trajectories per level). Table 2 details the Success Rate (SR) [%] and State Entropy (SE) of the demonstrated trajectories, alongside the networks' performance on train (square and star) and test shapes (plus and ellipse). The table shows several interesting results:

1. Introducing more noise increases the State Entropy (SE), but a severe noise corruption ($P = 200$) impairs the expert, lowering the Success Rate (SR) of the demonstrations, and degrading the robot's performance.

2. Adding the "right" amount of noise ($P = 100$) helps generalize to unseen shapes. Although less effective than masking, it still significantly outperforms the traditional approach, i.e., without any blindfolding (0 noise).

3. This experiment demonstrates that even a general form of blindfold (non-task specific) may still be superior to the conventional imitation learning approach.

Table 2: Success Rate (SR) [%] and State Entropy (SE) for the various blindfold types: range of noise levels or SAM2 mask. The column "Demos" details the SR and SE of experts' demonstration subject to the blindfold. The left-most columns are their corresponding evaluation performance [%] on Train (square, star) and Test (plus, ellipse).

| Max-Noise Level | Demos | | Train | | Test | |
|---|---|---|---|---|---|---|
| | SR | SE | square | star | plus | ellipse |
| 0 | 100% | 3.17 | 71% | 67% | 20% | 17% |
| 100 | 96% | 3.26 | 100% | 88% | 96% | 79% |
| 170 | 95% | 3.39 | 92% | 71% | 58% | 62% |
| 200 | 58% | 3.46 | 46% | 38% | 21% | 54% |
| SAM2 Mask | 100% | 3.52 | 96% | 96% | 100% | 96% |

# 5 Related works

Our work closely relates to imitation learning, information bottleneck, and robotic manipulation.

**Imitation learning theory.** The imitation learning literature counts a plethora of contributions, for which we refer to recent surveys [19, 60]. Here we are concerned with the generalization of behavioral cloning – a dominant imitation learning technique. Previous works [41, 42, 56, 38, 36, 37, 50, 13] have studied the theoretical limits of behavior cloning and settled the generalization gap on the training task as $J(\pi^*) - J(\widehat{\pi}) \lesssim R \log(|\Pi|/\delta)/n$ [13]. These results are mostly limited to cloning a Markovian policy in a single MDP. Other works have considered more general settings, including cloning history-based policies [4] and imitation learning under partial observability [48]. However, the generalization of behavior cloning in a meta learning setup, which is popular in empirical works [11, 12], is understudied. The only generalization analysis that strikes close to this setting is, to the best of our knowledge, the one in [40]. Differently from ours, they assume online access to the set of tasks to further fine-tune the cloned policy and they do not study how the information available to the demonstrator affects generalization, which is our main theoretical contribution.

**Imitation learning in robotics.** Imitation learning has been fundamental to various robotic domains, including autonomous driving [32], locomotion [31], flight [1], and manipulation [12]. A recent survey on manipulation, our case of interest here, is provided in [2]. Several works showed that imitation is useful for learning complex visuo-motor policies for robotic manipulation, where key ideas include predicting a sequence of future actions, and using a diffusion generative model to learn a distribution over the action sequence [6, 59, 14]. However, it is known that imitation learning requires a large number of demonstrations in order to generalize to variations in the task. Previously studied mitigations include 3-dimensional priors in the representation [58], automatic data augmentation [26, 15, 57], interactive data collection [18], and using simulations [52]. Another approach is leveraging large-scale data, either by collecting diverse task variations [23], or by fine tuning robotics foundations models [30, 49, 21]. Differently from the approaches above, we postulate that the way a demonstrator performs a task affects generalization, showing that by blindfolding the demonstrator we obtain an exploratory behavior, which induces better generalization when cloned. The generalization of exploratory behavior has been demonstrated in zero-shot reinforcement learning [61]. In comparison, we apply this idea to imitation learning, which requires a different approach, and also develop a theoretical explanation for the improved generalization. The idea that exploration at test time helps generalization has also been explored in the sim-to-real context in [54].

**Information bottleneck.** When learning a $X \to Y$ relationship between random variables, the information bottleneck [51] prescribes to "squeeze" $X$ into a representation that only retains information to predict $Y$. This principle is believed to be a factor beyond the generalization capabilities of deep learning [44] and formal generalization bounds through the information bottleneck have been derived [45, 29, 20]. In imitation learning, the information bottleneck has been used to analyze generalization in [3]. However, they consider generalization on the training task only and the information bottleneck is applied to the representation of the cloned policy. Our work advocates for applying an information bottleneck to the demonstrator to improve generalization of the cloned policy.

# 6 Conclusion

We showed that cloning the behavior of blindfolded experts leads to better generalization to unseen tasks. We supported this with theoretical analysis and conducted empirical tests that, for the first time, explored the concept of blindfolding experts in the context of a real-world robotic task, as well as a maze videogame. We observed that in both peg insertion and maze-solving tasks, blindfolding the experts encouraged them to enact a more exploratory behavior, cloned to produce policies that better generalize. Importantly, our approach achieves better generalization while accommodating any imitation learning algorithm.

Finally, we point out a limitation of the proposed approach: each domain may require a different kind of blindfold (e.g., concealing the field of view in the maze videogame, or masking out the shape of the hole in the peg insertion task). Too little obstruction does not elicit exploration, while redacting too aggressively would impede any informative exploratory behavior (may resort to near random walk). An interesting question is how to find the optimal balance, which we reserve for future research.

## Acknowledgments

This research was Funded by the European Union (ERC, Bayes-RL, 101041250). Views and opinions expressed are however those of the author(s) only and do not necessarily reflect those of the European Union or the European Research Council Executive Agency (ERCEA). Neither the European Union nor the granting authority can be held responsible for them.

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

# A Proofs

Here we provide the derivations for the generalization bound in Section 3. The proof requires a non-trivial combination of previous results in provable imitation learning, mostly from [13], and the generalization guarantees of information bottleneck [20]. These results are reported in Lemma A.3 and A.4. Before going through the proofs, we state the main theorem again for convenience.

**Theorem A.1** (Theorem 3.1). *For a confidence $\delta \in (0,1)$, it holds with probability at least $1 - 2\delta$*

$$\mathbb{E}_{T \sim P_0}[J(\pi_T^*) - J(\widehat{\pi})]$$

$$\lesssim RH\left(\mathcal{E}_{gen}(\pi^E) + \mathcal{E}_{opt}(\widehat{\pi}) + C(E, \pi^E, \widehat{\pi})\sqrt{\frac{I_{T;Z}|\mathcal{A}|\log(|\mathcal{A}|/\delta)}{m}} + \frac{8\log(|\Pi|m/\delta)}{n}\right)$$

*where*

- $R$ *is an upper bound to the cumulative reward of any policy in any task $\theta \in \Theta$ (Section 2);*
- $\mathcal{E}_{gen}(\pi^E)$ *is the generalization error of the expert's policy (Asm. 2);*
- $\mathcal{E}_{opt}(\widehat{\pi})$ *is a bound to the optimization error of solving* (1) *(Asm. 4);*
- $C(E, \pi^E, \widehat{\pi})$ *is a constant that depends on the training data $E$, the expert's policy $\pi^E$, the cloned policy $\widehat{\pi}$, and other absolute constants as detailed in Lemma A.4.*

*Proof.* We derive the result as follows

$$\mathbb{E}_{T \sim P_0}[J(\pi_T^*) - J(\widehat{\pi})]$$

$$\leq RH \mathbb{E}_{T \sim P_0} \mathbb{E}_{\tau \sim \mathbb{P}_T^{\pi^*}}[\mathbf{1}(\pi_T^*(a^h|x^h) \neq \widehat{\pi}(a^h|x^h))] \tag{4}$$

$$\leq RH\mathcal{E}_{gen}(\widehat{\pi}) \tag{5}$$

$$\leq RH\left(\mathcal{E}_{gen}(\pi^E) + \mathbb{E}_{T \sim P_0} \mathbb{E}_{XA \sim \mathbb{P}_T^{\pi^E}}[\mathbf{1}(\widehat{\pi}(X) \neq A)]\right) \tag{6}$$

$$\lesssim RH\left(\mathcal{E}_{gen}(\pi^E) + \mathcal{E}_{opt}(\widehat{\pi}) + \sqrt{\frac{I_{T;Z}|\mathcal{A}|\log(|\mathcal{A}|/\delta)}{m}} + \frac{8\log(|\Pi|m/\delta)}{n}\right) \tag{7}$$

where (4) and (5) are straightforward from the definitions of the performance $J(\pi)$ and the generalization error $\mathcal{E}_{gen}(\pi)$ (see Section 2 and (3) respectively), (6) follows from Assumption 2, and (7) holds with probability at least $1 - 2\delta$ through Lemma A.2. $\square$

We provide below the lemmas we need to prove the result above.

**Lemma A.2.** *For a confidence $\delta \in (0,1)$, it holds with probability at least $1 - 2\delta$*

$$\mathbb{E}_{T \sim P_0} \mathbb{E}_{XA \sim \mathbb{P}_T^{\pi^E}}[\mathbf{1}(\widehat{\pi}(X) \neq A)] \lesssim \sqrt{\frac{I_{T;Z}|\mathcal{A}|\log(|\mathcal{A}|/\delta)}{m}} + \frac{8\log(|\Pi|m/\delta)}{n} + \mathcal{E}_{opt}(\widehat{\pi})$$

*where $I_{T;Z}$ is the mutual information between the task $T$ and the internal representation of the demonstrator $Z$.*

*Proof.* We derive the result as follows

$$\mathbb{E}_{T \sim P_0} \mathbb{E}_{XA \sim \mathbb{P}_T^{\pi^E}}[\mathbf{1}(\widehat{\pi}(X) \neq A)]$$

$$\leq \mathbb{E}_{T \sim P_0} \mathbb{E}_{XA \sim \mathbb{P}_T^{\pi^E}}[\mathbf{1}(\widehat{\pi}(X) \neq A)] - \frac{1}{mnH}\sum_{i=1}^m \sum_{j=1}^n \sum_{h=0}^{H-1} \mathbf{1}(\widehat{\pi}(\tau_{ij}^h) \neq a_{ij}^h) + \mathcal{E}_{opt}(\widehat{\pi}) \tag{8}$$

$$\leq \mathbb{E}_{T \sim P_0} \mathbb{E}_{XA \sim \mathbb{P}_T^{\pi^E}}[\mathbf{1}(\widehat{\pi}(X) \neq A)] - \frac{1}{m}\sum_{i=1}^m \mathbb{E}_{XA \sim \mathbb{P}_{\theta_i}^{\pi^E}}[\mathbf{1}(\widehat{\pi}(X) \neq A)] + \frac{8\log(|\Pi|m/\delta)}{n} + \mathcal{E}_{opt}(\widehat{\pi}) \tag{9}$$

$$\lesssim \sqrt{\frac{I_{T;Z}|\mathcal{A}|\log(|\mathcal{A}/\delta)}{m}} + \frac{8\log(|\Pi|m/\delta)}{n} + \mathcal{E}_{opt}(\widehat{\pi}) \tag{10}$$

where (8) is a trivial consequence of Assumption 4 on the optimization error for solving (1), (9) holds with probability at least $1 - \delta$ through Lemma A.3 and a union bound on the $m$ training tasks, and (10) holds with probability $1 - 2\delta$ from Lemma A.4 by omitting constant and lower order terms and applying a union bound. $\qquad\square$

**Lemma A.3** (Sample complexity of behavioral cloning [13]). *For a confidence $\delta \in (0, 1)$, an MDP $\theta$, and a deterministic expert's policy $\pi^E$, it holds with probability at least $1 - \delta$*

$$\mathop{\mathbb{E}}_{XA\sim\mathbb{P}_\theta^{\pi^E}}[\mathbf{1}(\widehat{\pi}(X) \neq A)] \leq \frac{8\log(|\Pi|/\delta)}{n}.$$

*Proof.* This result can be obtained through a combination of results in [13]. First, for a policy $\widehat{\pi}$ obtained by minimizing the negative log likelihood of the data, as in (1), from Proposition 2.1 [13] we have with probability at least $1 - \delta$ that

$$D_H^2(\mathbb{P}_\theta^{\widehat{\pi}}, \mathbb{P}_\theta^{\pi^E}) \leq \frac{2\log(|\Pi|/\delta)}{n}$$

where $D_H^2(\mathbb{P}, \mathbb{Q}) = \int \left(\sqrt{d\mathbb{P}} - \sqrt{d\mathbb{Q}}\right)^2$ is the squared Hellinger distance between the probability measures $\mathbb{P}$ and $\mathbb{Q}$. Then, through Lemma F.3 [13] we have

$$\mathop{\mathbb{E}}_{XA\sim\mathbb{P}_\theta^{\pi^E}}[\mathbf{1}(\widehat{\pi}(X) \neq A)] \leq 4D_H^2(\mathbb{P}_\theta^{\widehat{\pi}}, \mathbb{P}_\theta^{\pi^E})$$

which concludes the proof. $\qquad\square$

**Lemma A.4** (Information bottleneck generalization gap [20]). *For a dataset $E = \{\theta_i \sim P_0\}_{i=1}^m$ of $m$ tasks and a single-point convex loss $\ell(\widehat{\pi}(X), A)$, let us define the generalization gap across the prior $P_0$ as*

$$\Gamma(E) := \mathop{\mathbb{E}}_{T\sim P_0} \mathop{\mathbb{E}}_{XA\sim\mathbb{P}_T^{\pi^E}}[\ell(\widehat{\pi}(X), A)] - \frac{1}{m}\sum_{i=1}^m \mathop{\mathbb{E}}_{XA\sim\mathbb{P}_{\theta_i}^{\pi^E}}[\ell(\widehat{\pi}(X), A)].$$

*For a confidence $\delta \in (0, 1)$, $\Gamma(E)$ is upper bounded with probability at least $1 - \delta$ by*

$$\beta\sqrt{\frac{I_{T;Z|A}\log 2 + \alpha_\gamma\log 2 + H(Z|T, A) + \log(2|\mathcal{A}|/\delta)}{m}} + \frac{f(\widehat{\pi})\sqrt{2\gamma|\mathcal{A}|\log(2|\mathcal{A}|/\delta)}}{m^{3/4}} + \frac{\gamma g(\widehat{\pi})}{m^{1/2}}$$

*where $\alpha_\gamma, \beta, \gamma$ are constant values, $f(\widehat{\pi}) = \max_{i\in[m]} \mathbb{E}_{XA\sim\mathbb{P}_{\theta_i}^{\pi^E}}[\ell(\widehat{\pi}(X), A)]$ is the maximum training loss, $g(\widehat{\pi}) = \sup_{XA} \ell(\widehat{\pi}(X), A)$ is the maximum generalization loss, and $H(Z|T, A)$ is the entropy of the demonstrator internal representation given $TA$.*

*Proof.* This result is based on Theorem 1 in [20], in which the notation adapted to our setting of interest. All of the derivations can be found in [20]. A coarser version of the bound is given as

$$\Gamma(E) \leq \left(\beta\sqrt{\alpha_\gamma\log 2 + H(Z|T, A)} + f(\widehat{\pi})\sqrt{2\gamma} + \gamma g(\widehat{\pi})\right)\sqrt{\frac{I_{T;Z|A}|\mathcal{A}|\log(2|\mathcal{A}|/\delta)}{m}}$$

where the first factor can be incorporated into a constant $C(E, \pi^E, \widehat{\pi})$. We note that the term $H(Z|T, A) = 0$ whenever the demonstrator internal representation is deterministic, which is a fair assumption in our setting. Further, the maximum training error $f(\widehat{\pi})$ is close to zero and upper bounded by the optimization error $\mathcal{E}_{opt}(\widehat{\pi})$. The value of $g(\widehat{\pi})$ is upper bounded by 1 for the indicator loss $\ell(\widehat{\pi}(X), A) = \mathbf{1}(\widehat{\pi}(X) \neq A)$. Finally, we note that $I_{T;Z} \geq I_{T;Z|A}$. With these considerations, by omitting all of the constants, we have

$$\Gamma(E) \lesssim \sqrt{\frac{I_{T;Z}|\mathcal{A}|\log(|\mathcal{A}|/\delta)}{m}}$$

as it is reported elsewhere in the paper. $\qquad\square$

# B Peg insertion extended results

This section describes in detail the setting, hyperparameters, and constants used for the peg insertion task, as well as provides extended evaluation results.

## B.1 Experimental setup

Figure 7 shows a close-up view of all peg shapes (10 shapes in total) and their corresponding boards, where the training shapes are in the top row and the test shapes are in the bottom row. Each peg insertion attempt starts from $10cm$ above the hole (Z axis) and a random reset position within a $0.5cm$ box in the XY plane, centered above the hole position. In this experiment, initial rotations about the X and Y axes are fixed (0 degrees), while the Z angle starts from a random rotation ranging from $-60$ to 60 degrees. Two Realsense cameras are mounted on the robot's wrist. For the blindfolded expert experiment, we mask out the hole to hide its orientation from the experts, such that they cannot infer the orientation of the peg and must explore the domain in order to insert the peg.

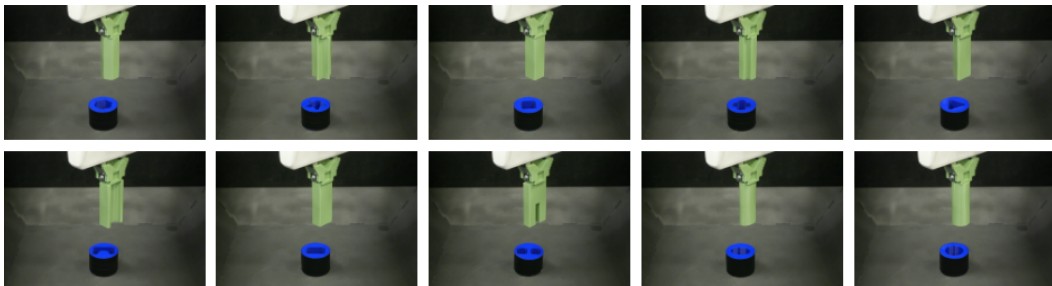

Figure 7: Close-up view of all peg insertion tasks. Top row: Training pegs. Bottom row: Test pegs.

## B.2 Hyperparameters and constants

As described in section 4.2, the same architecture is used for learning both $\pi_{BC}$ and $\pi_{BF-BC}$. Specifically, we use ResNet-10 [17] encoder pretrained on the ImageNet dataset [10], and a GRU of 1024, which we found to produce the best performance for both policies $\pi_{BC}$ and $\pi_{BF-BC}$ independently. Throughout our experiments, we train our networks using the Adam optimizer. The hyperparameters of the networks are the learning rate, learning rate decay, and the batch size. To ensure that the best performance of each approach is achieved, we perform a separate hyperparameter search for each policy, trained on each subset of shapes. The best hyperparameters are listed in Table 3. The network outputs a 6-dimensional vector for the mean and a 6-dimensional vector for the diagonal covariance matrix of a Gaussian policy (for 6-DoF action space). We train our networks using the log-likelihood loss. In all our evaluations, the action is chosen as the maximum likelihood of the distribution.

Table 3: List of hyperparameters used in the peg insertion experiment. The learning rate (lr) schedule indicates the iteration number for multiplying the lr by $0.5$.

| Hyperparameter | $\pi_{BC}$ | $\pi_{BF-BC}$ |
|---|---|---|
| batch size | 1024 | 1024 |
| hidden size | 1024 | 1024 |
| initial lr | 0.0003 | 0.0003 |
| lr schedule | lr $\times 0.5$ at $\{10, 100, 150, 200\}K$ | lr $\times 0.5$ at $\{50, 100, 150, 200\}K$ |

## B.3 Measuring exploratory behavior

We compute two additional measures for the exploratory behavior of the different experts: the map coverage score (Table 4) and the entropy of state visitation (Table 5).

**Map coverage score** is the ratio $C = N_v/N_{total}$ given by the number of visited states $N_v$ divided by all accessible states $N_{total}$, averaged over all episodes. For the peg insertion experiment, we consider

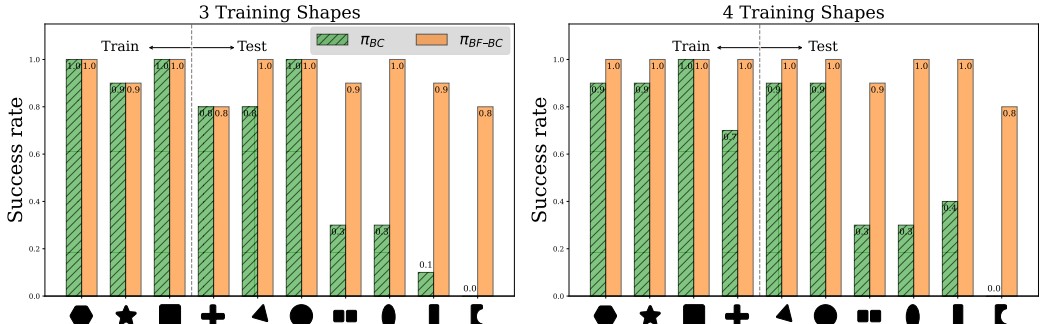

Figure 8: Success rate of robotic peg insertion for 10 peg shapes (horizontal axis). We train on a subset of shapes $k = \{3, 4\}$, with the remaining shapes withheld as test-set. Both for $k = 3$ (left) and $k = 4$ (right), cloning blindfolded experts generalizes better than cloning standard experts.

the rotation of the robotic arm around the Z-axis as the crucial component of the state space for obtaining the correct articulation for insertion. We compute the ratio of the rotation performed (in radians) divided by $2\pi$ in each trajectory, averaged over all trajectories.

**The entropy of state visitation** is defined by $H = -\sum_s p(s) \log p(s)$. We calculate $p(s)$ using a histogram (with 20 bins) of rotation angles along the trajectory, averaged over all trajectories.

The results confirm that blindfolded experts explore a larger portion of the state space to compensate for the redacted information in the observations.

Table 4: Map coverage score of the trajectories demonstrated by fully-informed experts (Experts) and blindfolded experts (BF-Experts).

| Mode | hexagon | star | square | plus | triangle | Average |
|------|---------|------|--------|------|----------|---------|
| Experts | 0.078 | 0.113 | 0.150 | 0.153 | 0.191 | 0.137 |
| BF-Experts | 0.114 | 0.259 | 0.267 | 0.270 | 0.327 | 0.247 |

Table 5: Entropy of the state visitation of the trajectories demonstrated by fully-informed experts (Experts) and blindfolded experts (BF-Experts).

| Mode | hexagon | star | square | plus | triangle | Average |
|------|---------|------|--------|------|----------|---------|
| Experts | 2.876 | 3.121 | 3.266 | 3.228 | 3.418 | 3.182 |
| BF-Experts | 3.100 | 3.416 | 3.546 | 3.577 | 3.631 | 3.454 |

### B.4 Results for different combinations of training shapes

Figure 8 shows the success rate for $k = 3, 4$ training shapes (and the rest serve as a test set, out of a total of 10 peg shapes). The results on the varying amounts of training shapes, further support that cloning blindfolded experts generalizes better than the standard BC approach.

## C Procgen maze and heist extended results

This section describes in detail the hyperparameters and constants used for the Procgen maze task.

### C.1 Hyperparameters and constants

For a fair comparison, we conduct a separate hyperparameter search for both $\pi_{BC}$ and $\pi_{BF-BC}$. We perform a hyperparameter search for the batch size $b \in \{128, 256, 512, 1024\}$, for the learning rate $\mathrm{lr} \in \{1e^{-3}, 1e^{-4}, 1e^{-5}, 5e^{-3}, 5e^{-4}, 5e^{-5}\}$ and for hidden size $h \in \{128, 256, 512, 1024\}$. We also evaluate the performance with and without using a learning decay schedule. Our networks are trained using the Adam optimizer. The best hyperparameters are chosen based on the lowest training loss and the highest training success rate. We evaluate performance over an average of 10 different random

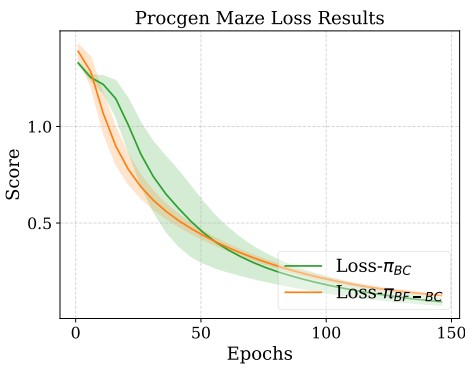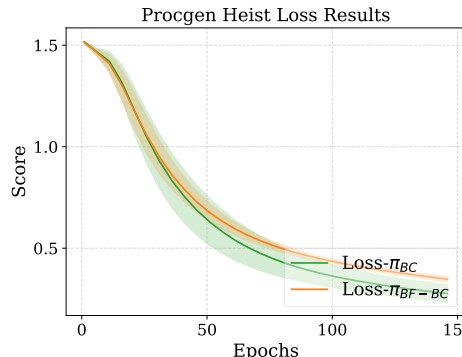

Figure 9: Loss as a function of training epochs for the Procgen maze (left) and heist (right). Mean (color line) and std (shaded region) are computed across 10 seeds.

training seeds. Figure 9 and Table 6 show the loss function and the chosen hyperparameters. Note that in most cases, the best hyperparameters for $\pi_{BC}$ and $\pi_{BF-BC}$ turned out to be fairly similar.

Table 6: List of hyperparameters used in the Procgen Maze experiment.

| Hyperparameter | $\pi_{BC}$ | $\pi_{BF-BC}$ |
|---|---|---|
| batch size | 256 | 256 |
| hidden size | 1024 | 1024 |
| learning rate | $10^{-5}$ | $10^{-5}$ |
| learning rate schedule? | No | No |

Table 7: List of hyperparameters used in the Procgen Heist experiment.

| Hyperparameter | $\pi_{BC}$ | $\pi_{BF-BC}$ |
|---|---|---|
| batch size | 128 | 128 |
| hidden size | 512 | 1024 |
| learning rate | $10^{-5}$ | $10^{-5}$ |
| learning rate schedule? | No | No |

## C.2 Number of steps vs. number of trajectories

As described in Table 1, the blindfolded expert takes more steps on average to complete each trajectory. When comparing the different approaches, we match the number of trajectories, which leads to a greater total number of environment steps for the blindfolded expert. Table 8 shows the total number of steps available for training the different BC policies, alongside their performance. We also compare our results to a standard BC approach with twice the number of trajectories (from the same 100 seeds) to match the number of environment steps produced by the blindfolded expert. In addition, we compare our results to the results reported by [27], who train a BC policy on the Procgen maze with a dataset of $1M$ environment steps taken from a trained PPO expert, on 200 training seeds[8].

We can see in Table 8 that $\pi_{BF-BC}$ achieves better performance than all other contending policies. When compared to the results reported in [27], we can see that our results are better despite significantly less training data (an order of magnitude fewer trajectories) and half the number of training seeds.

---

[8]For $1M$ expert dataset, we report the results from [27] who evaluated over 5 seeds.

Table 8: Performance comparison on the Procgen maze experiment. Our results are reported at epoch 40 (early stopping) when training performance plateaus. The mean and std are computed over 10 seeds. Top performer in bold.

| Parameter | 1M Expert Dataset in [27] | $\pi_{BC}$ | $\pi_{BC-ext}$ | $\pi_{BF-BC}$ |
|---|---|---|---|---|
| # of trajectories | 15385 | 2000 | 4608 | 2000 |
| # of total env steps | 1000000 | 57166 | 115290 | 103238 |
| # of seeds | 200 | 100 | 100 | 100 |
| Test performance | $4.46 \pm 0.16$ | $3.35 \pm 0.29$ | $3.65 \pm 0.3$ | $\mathbf{6.37 \pm 0.47}$ |

# D  Data collection

**Peg insertion.**      For both $\pi_{BC}$ and $\pi_{BF-BC}$, we use $400$ trajectories for each of the training shapes. We, the authors, collected the data by operating the robot manually using a Spacemouse control. Recall that the blindfolded expert observes a masked-out view of the board (through the robot wrist cameras) such that the orientation of the peg is not directly visible and must be inferred through exploration. However, recorded observations in favor of cloning the blindfolded policy $\pi_{BF-BC}$ are unmasked, i.e., only the human expert is blindfolded. In addition, we rescale the images from $480 \times 480$ to $128 \times 128$ for both $\pi_{BC}$ and $\pi_{BF-BC}$ to facilitate computations.

**Procgen maze and heist.**

To train our experts (BC) and blindfolded experts (BF-BC) policies, we collected $4000$ human demonstrations on 200 levels. We conducted crowd-sourced data collection for the maze and heist videogames by recruiting 20 volunteers who played the games. Each expert played all 200 levels of maze and all 200 levels of heist twice, once with the mask and once without the mask. Their game trajectories were recorded to serve toward the imitation-learning of the experts' policy $\pi_{BC}$ and blindfolded experts' policy $\pi_{BF-BC}$. The participants moved the mouse using the keyboard's arrow keys and relied on the Procgen "interactive" GUI for maze and heist observations in full resolution ($512 \times 512$). For the blindfolded experts' data, their observations are modified to reveal only the agent's immediate surroundings (a diameter of $\frac{1}{8}$ of the width for maze and $\frac{1}{6}$ for heist) with the rest of the observation masked out. Note that the state observations that are provided to the cloning networks are a lower resolution of $64 \times 64$ of the unmasked observations.

All participants were compensated with vouchers for their efforts. The experiment's GUI environment, alongside the training code and the recorded data, are available at:
`https://github.com/EvZissel/blindfolded-experts/`

