# OpenReview forum: "Blindfolded Experts Generalize Better:  Insights from Robotic Manipulation and Videogames"
_NeurIPS.cc/2025/Conference — NeurIPS 2025 poster_

### Official Review · Reviewer_qfwW · 2025-06-10

**Clarity:** 4
**Significance:** 4
**Originality:** 4
**Rating:** 5
**Confidence:** 5

**Summary:**

This paper introduces a novel and insightful perspective on imitation learning: the quality and nature of expert demonstrations are as critical for generalization as their quantity. The authors propose the concept of a "blindfolded" expert—an expert who is intentionally deprived of some task-specific information. This forces the expert to adopt a more general, exploratory behavior to solve the task. The core contribution is showing, both theoretically and empirically, that cloning this exploratory behavior leads to policies that generalize significantly better to unseen tasks. This central idea is supported by a formal generalization bound, which links the error to the mutual information between the expert's strategy and the task identity, and is validated with compelling experiments in a Procgen maze environment and on a real-world robotic peg insertion task.

**Questions:**

- Could the authors comment on the sensitivity of the method to the degree of "blindfolding"? The experiments show a case that works well, but what happens at the extremes? For instance, if the blindfold is so severe that the expert's success rate drops significantly, how does this affect the performance of the cloned policy? This speaks to the practical challenge of designing a good blindfold.
- The choice of "blindfold" seems domain-specific (hiding the global maze vs. masking the hole shape). How sensitive were the results to the specific implementation of the mask? For the peg insertion task, for example, what would happen if instead of masking the hole, you added significant noise to the robot's proprioceptive state, forcing the expert to rely more on visual and force feedback to re-localize the peg? This could help understand if the benefit comes from any information bottleneck, or specifically from withholding task-identity information.

**Ethical Concerns:**

["NO or VERY MINOR ethics concerns only"]

**Final Justification:**

This is an excellent paper that shed new light on the relationship between the demonstration data and the resulting imitation learning policy, which should be of great interest to the robot learning community.

**Limitations:**

The experiments are conducted on tasks that, while challenging, have a relatively monolithic goal (find the cheese, insert the peg). In more complex, sequential tasks (e.g., "pick up the cup, take it to the sink, turn on the faucet"), it's unclear how to apply a meaningful blindfold. Masking the final goal might make the task entirely unsolvable, while masking an intermediate step might not induce the right kind of hierarchical exploration. The proposed blindfolding strategy may not scale intuitively to tasks with long-horizon, compositional logic.

**Quality:**

4

**Strengths And Weaknesses:**

Strengths:
+ The paper provides a refreshing and valuable lens on data collection for imitation learning. The community has largely focused on scaling data quantity, using data augmentation, or improving the learning algorithm itself. This work compellingly argues that we must also consider the behavioral patterns within the demonstrations. The concept of "blindfolding" to induce a more generalizable, exploratory strategy is both intuitive and powerful. This has immediate practical implications for robotics and other fields where collecting vast amounts of "perfect" demonstrations is prohibitively expensive.

Weaknesses:
- A primary limitation lies in the abstract nature of the expert's internal representation, Z, and the associated mutual information term, I(Z;T). This term is central to the theoretical argument, but it is not practically measurable. The authors make a reasonable assumption that masking the expert's input observation will reduce I(Z;T), but this is an indirect manipulation. The paper would benefit from a discussion on this gap, acknowledging that Z is an unobservable theoretical construct and the link to input masking is an assumption, albeit a well-motivated one.
- Furthermore, the paper does not offer a principle for designing the optimal "blindfold." There is a clear trade-off: too little masking may not induce exploration, while too much masking may render the task unsolvable for the expert, leading to poor or failed demonstrations. The success of the method hinges on finding a sweet spot, and the paper currently lacks guidance on how a practitioner might find this balance for a new task.

---

> ### Author Rebuttal · Authors · 2025-07-31
>
> We thank the reviewer for the insightful comments. We are pleased that our work is found to be "intuitive and powerful" and recognized for its "immediate practical implications for robotics and other fields". In the following, we provide detailed replies to the questions and concerns.
>
> **How to quantify the mutual information I(Z;T)?**
>
> This is a very good point. We can still lower bound the information between *observations* and *tasks* using a variational approach -- a classifier that predicts task identity from the observations [1]. We believe that this bound will be lower when we add the mask, and may be used in the future to devise better masks. Of course, whatever information the human gleans from the observation is not as easy to quantify.
>
> **How to design the blindfold?**
>
> The study of principled methods for automatic design of the blindfold is an interesting direction for future work.
>
> Here we can provide some insights on how we designed the blindfold in practice. Take the peg insertion case, in which the blindfold is a mask on the image observation. We tweaked the mask to hide most information that still allows the human to solve the task. At that point, removing even more information would not have produced meaningful demonstrations.
> This heuristic can be replicated by practitioners in other domains.
>
> **Question 1.**
>
> We expect the sensitivity to the degree of blindfolding to follow the theoretical result in Theorem 3.1: More blindfolding leads to more generalization. However, if the blindfold is so severe that the expert's success rate drops, we do not expect the cloned policy to work well. Recall that we are doing behavioral cloning (and not offline RL), so the performance of the policy is limited by the performance of the expert.
>
> **Question 2.**
>
> We tested a single implementation of the mask, so we do not have a sensitivity analysis.
> The idea of adding noise is interesting for this matter: We can add experiments with different noise added to the observations (proprioceptive noise is less relevant as the demonstrator can see the robot in the image, but we can add image noise instead of masking the hole).
>
> **Address limitation: Beyond monolithic goals.**
>
> We acknowledge that future studies are required to determine the full potential of blindfolding. That being said, our method is not a silver bullet for all generalization problems.
>
> Nevertheless, we provide an additional experiment with some glimpses of the challenges the reviewer is suggesting.
> In Procgen Heist, the agent has to solve a sequence of sub-goals in order to solve the task. At each level, the agent needs to collect a gem hidden behind a network of color-coded locks accessible through keys (of corresponding colors) that are scattered throughout the level (See [1] for more details on the environment setting). This task requires multi-stage planning in the form of lock-waypoints and to overcome variations in sub-task ordering, making the task more complex and challenging than the regular maze. For training the BC policies, we collected human demonstrations as in the Maze experiment, applying a similar blindfold to the blindfolded experts. In Table 1 below, we compare the results of the agent trained with the fully informed expert's data ($\pi_{BC}$) to the results of the agent trained on the blindfolded experts' data ($\pi_{BF-BC}$). While the training performance is above 90% for both agents, $\pi_{BF-BC}$ significantly outperforms $\pi_{BC}$ on unseen test levels (43% vs. 7%), which indicates better generalization performance (results obtained over $4K$ trajectories from $200$ different seeds).
>
> *Table 1: Performance comparison on the Procgen Heist experiment. The mean and std are computed over 10 seeds. Top performer in bold.*
> | Method | Train | Test |
> |--------|-------|------|
> | $\pi_{BC}$ | 9.41 ± 0.20 | 0.69 ± 0.44 |
> | $\pi_{BF-BC}$ | **9.91 ± 0.04** | **4.27 ± 1.01** |
>
> **References**
>
> [1] Cobbe, Karl, et al. "Leveraging procedural generation to benchmark reinforcement learning." International conference on machine learning. PMLR, 2020.

---

> > ### Comment · Reviewer_qfwW · 2025-08-05
> >
> > Thanks for the detailed response. This is a great paper and I'll keep my rating as "Accept"

---

> > > ### Author Response · Authors · 2025-08-08
> > > **Grateful for your insights**
> > >
> > > Thank you for the great feedback!
> > >
> > > Following your suggestion regarding the sensitivity to the choice of a blindfold, we conducted an experiment of robotic peg insertion, where we applied random noise to the experts' observations instead of masking the hole (random noise $p \sim U[0,P]$ is added to all pixel values).
> > > We collected a total of 1600 expert trajectories from varying maximum noise level $P \in [0, 100, 170, 200]$ and 2 training shapes (square and star), and trained a network on each noise level $P$ (i.e., 400 trajectories per level). The table below details the Success Rate (SR) [%] and State Entropy (SE) of the demonstrated trajectories, alongside the networks' performance on train (square and star) and test shapes (plus and ellipse).
> > > The table shows an interesting result:
> > > 1) Introducing more noise increases the State Entropy (SE), but a severe noise level (200) impairs the expert, lowering the Success Rate (SR) of the demonstrations, and degrading the robot's performance.
> > > 2) Adding the right amount of noise (100) helps generalize to unseen shapes. Although less effective than masking, it still significantly outperforms the traditional approach without any "blindfolding" (0 noise).
> > >
> > > This experiment further strengthens the generality of our approach - that even a more general form of blindfold is still superior to the conventional approach.
> > > As you pointed out, the concept of "blindfolding" to induce a more generalizable, exploratory strategy is both intuitive and powerful and has immediate practical implications for robotics and other fields. Your insights helped make this paper better and we're very grateful.
> > >
> > >
> > >
> > > **Table:** Success Rate (SR) [%] and State Entropy (SE) of the experts' demonstrations, and their corresponding evaluation performance [%] on Train (square, star) and Test (plus, ellipse) shapes for various noise levels.
> > >
> > > | Max-Noise Level | Demonstrations |     | Train |      | Test |         |
> > > |-----------------|-----------|---------|-------|------|------|---------|
> > > |                 | SR          | SE  | square| star | plus | ellipse |
> > > | 0               | 100%        | 3.17| 71%   | 67%  | 20%  | 17%     |
> > > | 100             | **96%**   | **3.26** | **100%** | **88%** | **96%** | **79%** |
> > > | 170             | 95%       | 3.39| 92%   | 71%  | 58%  | 62%     |
> > > | 200             | 58%       | 3.46| 46%   | 38%  | 21%  | 54%     |
> > > | SAM2 Mask       | **100%**    | **3.52** | **96%** | **96%** | **100%** | **96%** |

---

### Official Review · Reviewer_29eA · 2025-06-28

**Clarity:** 2
**Significance:** 2
**Originality:** 3
**Rating:** 4
**Confidence:** 3

**Summary:**

This paper presents a simple yet effective paradigm for behavioral cloning (BC). During demonstration data collection, the approach restricts the information available to human experts (observation masking), forcing them to adopt more exploratory behaviors to complete tasks. Crucially, while the expert operates with limited information, the recorded observations remain complete and unmasked. When these demonstrations are used for behavioral cloning, the learned policy inherits the expert's exploratory behavior. This data collection methodology significantly improves the generalization performance of the resulting policies.

**Questions:**

- How might you automatically determine the optimal "blindfolding" strategy for a new domain without domain-specific engineering?
- How does this approach compare to simply collecting more diverse demonstrations from standard experts? Is there a point where additional task diversity would make the blindfolding unnecessary?

**Ethical Concerns:**

["NO or VERY MINOR ethics concerns only"]

**Final Justification:**

The authors' rebuttal has addressed several of my concerns; however, I will maintain my original score due to the limitations regarding task-specific blindfolding.

**Limitations:**

yes

**Quality:**

2

**Strengths And Weaknesses:**

Strengths:
- Simple but effective way for improving BC: The idea of manipulating the expert's information access rather than modifying the learning algorithm is effective and orthogonal to existing approaches for improving BC.
- Solid theoretical analysis: The paper provides formal analysis showing why blindfolding experts leads to better generalization, connecting information theory to multi-task generalization.
- Good results: The experiments on both simulated and real-world tasks show substantial improvements in generalization performance, especially when trained on fewer task variations.

Weaknesses:
- Task-specific blindfolding: As acknowledged by the authors, each task requires a custom blindfolding strategy, and finding the right balance of information restriction is non-trivial.
- Limited task diversity: While the experiments cover both virtual and physical domains, they're limited to navigation and insertion tasks. It's unclear how the approach would generalize to more complex or diverse task families.
- Increased demonstration effort: Blindfolded experts take longer to complete tasks (as shown in Table 1), which means more time is required for data collection, potentially offsetting some of the data efficiency gains.

---

> ### Author Rebuttal · Authors · 2025-07-31
>
> We thank the reviewer for the insightful comments and for considering our approach to be effective and well-founded.
> We are reporting below detailed replies to the questions and concerns.
>
> **Task diversity.**
>
> We include additional results on Procgen Heist to increase the task diversity in the paper. In Heist, the agent needs to collect a gem hidden behind a network of color-coded locks accessible through keys (of corresponding colors) that are scattered throughout the level (See [1] for more details on the environment setting). This task requires multi-stage planning in the form of lock-waypoints and to overcome variations in sub-task ordering, making it significantly more complex than the regular maze. For training the BC policies, we collected human demonstrations as in the Maze experiment, applying a similar blindfold to the blindfolded experts. In Table 1 below, we compare the results of the agent trained with the fully informed expert's data ($\pi_{BC}$) to the results of the agent trained on the blindfolded experts' data ($\pi_{BF-BC}$). While the training performance is above 90% for both agents, $\pi_{BF-BC}$ significantly outperforms $\pi_{BC}$ on unseen test levels (43% vs. 7%), which indicates better generalization performance (results obtained over $4K$ trajectories from $200$ different seeds).
>
> *Table 1: Performance comparison on the Procgen Heist experiment. The mean and std are computed over 10 seeds. Top performer in bold.*
> | Method | Train | Test |
> |--------|-------|------|
> | $\pi_{BC}$ | 9.41 ± 0.20 | 0.69 ± 0.44 |
> | $\pi_{BF-BC}$ | **9.91 ± 0.04** | **4.27 ± 1.01** |
>
> **How the approach generalizes to more complex or diverse task families.**
>
> For robotic applications, a simple "rule of thumb" we found is -- can you solve it in the dark? Many important tasks fall under this criterion, including grasping, assembly, and disassembly. This is not exhaustive, and additional blindfold techniques may be found in future studies to extend the applicability of the idea.
>
> On this line, we mention an interesting (yet speculative) application to LLM reasoning. When answering a complex question (e.g., "what is the tallest tower in Europe"), we must balance between recalling from memory, and performing some reasoning computation ("what are the countries in Europe, what is the tallest tower in each European country, etc."). For supervised fine tuning, if we manage to induce the humans to provide more reasoning traces, they may lead to better generalization. We may "blindfold" the humans by asking them not to rely on memory, for instance, but more on tool use and reasoning.
>
> **Increased demonstration effort offsetting data efficiency gains**
>
> This is a valid point, for which we dedicated an ablation study in the manuscript (Figure 3). Since the blindfolded experts' trajectories are longer, we collected more trajectories with fully-informed experts ($\pi_{BC-ext}$) to match the total number of samples of the blindfolded expert. The results are mostly the same: This means that the blindfolded approach generalizes better **with the same number of samples** and therefore the same time for collecting the data.
>
> **Q1. How might you automatically determine the optimal "blindfolding" strategy for a new domain without domain-specific engineering?**
>
> This is an interesting direction that would require substantial additional study, which we leave as future work.
>
> We can provide some insights on how we designed the blindfold in practice. Take the peg insertion case, in which the blindfold is a mask on the image observation. We tweaked the mask to hide most information that still allows the human to solve the task. At that point, removing even more information would not have produced meaningful demonstrations. This heuristic can be replicated by practitioners in other domains.
>
> The latter requires some engineering, but note that any BC experiment requires engineering anyway (of the data collection setup, the human demonstrator instruction, etc.), and we believe blindfolds will become another factor in this list.
>
> **Q2. How does this approach compare to simply collecting more diverse demonstrations from standard experts? Is there a point where additional task diversity would make the blindfolding unnecessary?**
>
> If we understand the reviewer's intention correctly, "more diverse demonstrations" refers to "demonstrations of more tasks". The intuition is correct: **When demonstrating more and more tasks, the gap between blindfolded and standard experts shrinks**. This is supported both by the theory and the experiments.
>
> The theory shows that the generalization error scales with $O (\sqrt{I_{Z;T} / m})$, where $m$ is the number of demonstrated tasks. When $m$ is large enough, the effect of a smaller $I_{Z;T}$ is reduced.
>
> The experiment on peg insertion shows that the generalization error of the policy cloned from regular demonstrations shrinks slightly when increasing the training tasks from 2 (Figure 5 left) to 5 (Figure 5 right). The gap with the blindfolded strategy remains significant, but we expect it to reduce with even more demonstrated tasks.
>
> **References**
>
> [1] Cobbe, Karl, et al. "Leveraging procedural generation to benchmark reinforcement learning." International conference on machine learning. PMLR, 2020.

---

> > ### Comment · Reviewer_29eA · 2025-08-05
> >
> > The authors' rebuttal has addressed several of my concerns; however, I will maintain my original score due to the limitations regarding task-specific blindfolding.

---

> > > ### Author Response · Authors · 2025-08-08
> > > **Deeply appreciate your inputs**
> > >
> > > We appreciate your feedback and valuable input on our work.
> > >
> > > To address your concern regarding the task-specific blindfold, we conducted an experiment of robotic peg insertion, where we applied random noise to the experts' observations instead of masking the hole (random noise $p \sim U[0,P]$ is added to all pixel values).
> > > We collected a total of 1600 expert trajectories from varying maximum noise level $P \in [0, 100, 170, 200]$ and 2 training shapes (square and star), and trained a network on each noise level $P$ (i.e., 400 trajectories per level). The table below details the Success Rate (SR) [%] and State Entropy (SE) of the demonstrated trajectories, alongside the networks' performance on train (square and star) and test shapes (plus and ellipse).
> > > The table shows an interesting result:
> > > 1) Introducing more noise increases the State Entropy (SE), but a severe noise level (200) impairs the expert, lowering the Success Rate (SR) of the demonstrations, and degrading the robot's performance.
> > > 2) Adding the right amount of noise (100) helps generalize to unseen shapes. Although less effective than masking, it still significantly outperforms the traditional approach without any "blindfolding" (0 noise).
> > >
> > >
> > > This experiment demonstrates that even a more general form of blindfold (non-task specific) is still superior to the conventional approach, and we believe this fulfils your expectations.
> > >
> > > **Table:** Success Rate (SR) [%] and State Entropy (SE) of the experts' demonstrations, and their corresponding evaluation performance [%] on Train (square, star) and Test (plus, ellipse) shapes for various noise levels.
> > >
> > > | Max-Noise Level | Demonstrations |     | Train |      | Test |         |
> > > |-----------------|----------------|-----|-------|------|------|---------|
> > > |                 | SR Demos      | SE  | square| star | plus | ellipse |
> > > | 0               | 100%           | 3.17| 71%   | 67%  | 20%  | 17%     |
> > > | 100             | **96%**        | **3.26** | **100%** | **88%** | **96%** | **79%** |
> > > | 170             | 95%            | 3.39| 92%   | 71%  | 58%  | 62%     |
> > > | 200             | 58%            | 3.46| 46%   | 38%  | 21%  | 54%     |
> > > | SAM2 Mask       | **100%**       | **3.52** | **96%** | **96%** | **100%** | **96%** |

---

### Official Review · Reviewer_LUkx · 2025-07-01

**Clarity:** 4
**Significance:** 4
**Originality:** 3
**Rating:** 5
**Confidence:** 3

**Summary:**

This paper studies the effect of information bottlenecks on the expert’s observation in behavior cloning (BC), with a conjecture that such bottlenecks yield better generalization of BC policies across multiple tasks. The paper provides a theoretical result proving that an upper-bound on the generalization error scales with the sum of two terms: the cost of suboptimal experts and the square-root of mutual-information between the task information and the expert’s internal representation of the task. This theory suggests that lower mutual information for the expert leads to more generalizable multi-task BC policies. The paper empirically verifies this theory in a simulated Procgen domain and a real-world robotic manipulation domain.

**Questions:**

In the proof of Theorem 3.1, the authors note that equation (6) follows from Assumption 2. Some elaboration would be appreciated since the transition from (5) to (6) was not trivial to me.

**Ethical Concerns:**

["NO or VERY MINOR ethics concerns only"]

**Final Justification:**

The rebuttals by authors have addressed most of my major concerns and questions that I raised in my original review. While the authors admit how to reduce task information in language-conditioned settings is still an open problem, such an acknowledgement of limitation should be rewarded rather than punished since it improves clarity of the paper. The new Procgen Heist experiment demonstrates strong generalizability of blindfolded experts. Thus, I have raised the clarity and the significance scores. I believe that the paper is solid.

**Limitations:**

yes

**Quality:**

3

**Strengths And Weaknesses:**

**Strengths:**
* Multi-task generalization of behavior cloning is an important open research question. The paper poses a very interesting conjecture that information bottlenecks for the expert demonstrator will improve the generalizability of BC agents. This conjecture is convincingly supported by both theoretical and empirical results.

* Overall, the experimental setup is well-designed. The ablation of $\pi_{\text{BC-ext}}$ is a nice addition to the performance comparison, since it effectively eliminates the possibility that the blind-folded policy generalized simply due to increased amount of data.

* Despite the inherent complexity of the theoretical development, the paper is well-written and is relatively straightforward to follow.

**Weaknesses:**
* The main theoretical results (Theorem 3.1) essentially illustrate a trade-off between the degree of suboptimality of the expert and the amount of mutual information. As the paper admits in the Conclusion section, finding the best trade-off between the two conflicting terms can be task-specific and challenging to achieve in practice.

* Related to the point above, it is difficult to quantify the mutual information $I_{Z,T}$ since we don't know the expert's internal representation $Z$. Therefore, specific means to reduce this mutual information have to resort to some heuristic choices (such as masking parts of observations, as done in this paper), rather than some principled guidelines to design effective interventions based on quantified mutual information.

* Although the empirical results are generally convincing, the variation of tasks in the empirical study is somewhat small. Specifically, each maze layout and background is considered a distinct task in the Procgen domain, and each peg shapes in the manipulation domain. In future work, it would be interesting to see more diversity across different axes of generalization, including spatial, object, and visual variations, as well as task-specification diversities for language-conditioned policies. There are some simulation environments specifically designed to test multi-task policies in the robotics domain (such as the Meta-World [1] and LIBERO [2] benchmarks, to name a few).

[1] YU, Tianhe, et al. Meta-world: A benchmark and evaluation for multi-task and meta reinforcement learning. In: Conference on robot learning. PMLR, 2020. p. 1094-1100.

[2] LIU, Bo, et al. Libero: Benchmarking knowledge transfer for lifelong robot learning. Advances in Neural Information Processing Systems, 2023, 36: 44776-44791.

---

> ### Author Rebuttal · Authors · 2025-07-31
>
> We appreciate the reviewer's constructive comments. We discuss the raised points below.
>
> **It is difficult to quantify the mutual information since we don't know the expert's internal representation Z.**
>
> This is a very good point. We can still lower bound the information between *observations* and *tasks* using a variational approach -- a classifier that predicts task identity from the observations [1]. We believe that this bound will be lower when we add the mask, and may be used in the future to devise better masks. Of course, whatever information the human gleans from the observation is not as easy to quantify.
>
> **Task diversities and more variation of tasks.**
>
> As we discuss in the paper, not all task variability can be improved by our method. Following Theorem 3.1, we need the blindfold to reduce task information, but still allow the human to solve the task. In robotics, a simple "rule of thumb" for such tasks is -- can you solve it in the dark? Many important tasks fall under this criterion, including grasping, assembly, and disassembly. The tasks in Meta-World and Libero do not -- the observation contains the task description itself -- what to do and where to move, and they are impossible to solve without a very detailed observation. How to reduce task information in such domains is still an open problem.
>
> In the meantime, **we provide an additional experiment on Procgen Heist game** to have more variation of tasks in the paper. In Heist, the agent needs to collect a gem hidden behind a network of color-coded locks accessible through keys (of corresponding colors) that are scattered throughout the level (See [2] for more details on the environment setting). This requires multi-stage planning in the form of lock-waypoints and to overcome variations in sub-task ordering, making it significantly more complex than the regular maze. For training the BC policies, we collected human demonstrations as in the Maze experiment, applying a similar blindfold to the blindfolded experts. In Table 1 below, we compare the results of the agent trained with the fully informed expert's data ($\pi_{BC}$) to the results of the agent trained on the blindfolded experts' data ($\pi_{BF-BC}$). While the training performance is above 90% for both agents, $\pi_{BF-BC}$ significantly outperforms $\pi_{BC}$ on unseen test levels (43% vs. 7%), which indicates better generalization performance (results obtained over $4K$ trajectories from $200$ different seeds).
>
> *Table 1: Performance comparison on the Procgen Heist experiment. The mean and std are computed over 10 seeds. Top performer in bold.*
> | Method | Train | Test |
> |--------|-------|------|
> | $\pi_{BC}$ | 9.41 ± 0.20 | 0.69 ± 0.44 |
> | $\pi_{BF-BC}$ | **9.91 ± 0.04** | **4.27 ± 1.01** |
>
>
> **Q1. Elaboration on the transition from (5) to (6) in Theorem 3.1.**
>
> The derivation in Eq. 6 is
> $$\mathcal{E}\_{gen} (\hat{\pi}) \leq \mathcal{E}\_{gen} (\pi^E)+ \mathbb{E}\_{T \sim P\_0} \mathbb{E}\_{XA \sim \mathbb{P}^{ \pi^E}\_{T}} [1 (\hat\pi (X) \neq A)].$$
> This is obtained by noting that the suboptimality of the cloned policy $\hat \pi$ cannot be larger than the suboptimality of the expert's policy $\pi^E$ plus a term that accounts for every time $\hat\pi$ is different from $\pi^E$. Then, Assumption 2 allows us to write the suboptimality of the expert's policy as $\mathcal{E}_{gen} (\pi^E)$.
>
> **References**
>
> [1] Chen et al., Infogan: Interpretable representation learning by information maximizing generative adversarial nets. NeurIPS 2016.
>
> [2] Cobbe, Karl, et al. "Leveraging procedural generation to benchmark reinforcement learning." International conference on machine learning. PMLR, 2020.

---

> ### Comment · Reviewer_LUkx · 2025-08-04
> **Acknowledgement of Rebuttal**
>
> I sincerely thank the authors for their thorough response. It has addressed most of my major concerns and questions that I raised in my original review.
>
> > We can still lower bound the information between observations and tasks using a variational approach -- a classifier that predicts task identity from the observations [1]. We believe that this bound will be lower when we add the mask, and may be used in the future to devise better masks.
>
> This variational approach would be a very interesting idea for future work.
>
> Regarding the authors' explanation of equation (6), my interpretation is as follows:
> $$\mathbb{1}(\hat{\pi}(X) \neq A)  \leq \mathbb{1}(\pi^E(X) \neq A) + \mathbb{1}(\hat{\pi}(X) \neq \pi^E(X))$$
> for any $X$ and $A$. I verfied myself that it is true, but I want to make sure my interpretation is correct.
>
> While the authors admit how to reduce task information in language-conditioned settings is still an open problem, such an acknowledgement of limitation should be rewarded rather than punished since it improves clarity of the paper. (Please discuss this point in the final version.) The new Procgen Heist experiment demonstrates strong generalizability of blindfolded experts in tasks that require long-horizon and complex interaction with the environment. Thus, I have raised the clarity and the significance scores.

---

> > ### Author Response · Authors · 2025-08-05
> > **Thank you!**
> >
> > We thank the reviewer for the additional comments. We will definitely add all the discussions, clarifications, and experiments raised during the review process to our final version.
> >
> > Regarding equation (6): your interpretation is perfectly correct and provides a different way to write this equation.

---

> > > ### Comment · Reviewer_LUkx · 2025-08-05
> > >
> > > Sounds good to me, thank you for the additional clarification.

---

> > > > ### Author Response · Authors · 2025-08-08
> > > > **One more thing**
> > > >
> > > > Thank you again for your feedback and valuable input on our work.
> > > >
> > > > Perhaps you'll be interested in an experiment we conducted following reviewer qfwW's suggestion. In the robotic peg insertion task, we applied random noise to the experts' observations instead of masking the hole - leading to increased state entropy and reduced success rate as more noise is introduced (results table included in the latest comment for reviewer qfwW).
> > > >
> > > > The results show that even such a general form of blindfold significantly outperforms the conventional approach (which doesn't use any "blindfold"), further strengthening the generality of our approach.

---

### Official Review · Reviewer_bJoN · 2025-07-01

**Clarity:** 3
**Significance:** 3
**Originality:** 3
**Rating:** 5
**Confidence:** 3

**Summary:**

In this paper, the authors propose a method called blindfolded expert, in which experts are intentionally provided with only partial information. As a result, the expert cannot fully grasp the given task, prompting more exploratory behavior. The central hypothesis is that this exploratory behavior enables better generalization to previously unseen tasks.
Building on this hypothesis, the authors conduct a theoretical analysis and derive a relationship showing that the generalization error depends on the amount of task-specific information available to the expert. The empirical results show that this hypothesis through experiments on a real-world robotic peg insertion task and a video game environment (Procgen Maze).

**Questions:**

The strength of the information bottleneck (i.e., the degree of blindfolding) must be carefully tuned to encourage exploration while still allowing task success. This appears to require domain-specific tuning. Do the authors have any principled method or insight for finding this optimal trade-off?

**Ethical Concerns:**

["NO or VERY MINOR ethics concerns only"]

**Final Justification:**

The authors provides quantitative results regarding the exploration of the blindfolded expert. I believe this paper is solid and novel overall. Also, I encourage the authors to include a more explicit discussion of the limitations, especially the restriction to discrete action spaces in theoretical analysis. Also, it would be interesting how much this technique gives additional gain in different levels of difficulty.

**Limitations:**

The theoretical bound is derived under a classification setting and may not hold in continuous action spaces. Also, strong bottleneck could degrade expert performance. (the authors do acknowledge both of these limitations in the manuscript).

**Paper Formatting Concerns:**

no formatting concerns

**Quality:**

3

**Strengths And Weaknesses:**

Strengths

- Behavior cloning is an active area of research in robotics and decision making. While much work has focused on scalability with respect to the number of tasks or demonstrations, this paper presents a novel direction by investigating how performance scales with the mutual information between the expert’s internal representation and the task.

- In the case of the blindfolded expert, the trajectories tend to have a higher average number of steps. The authors acknowledge that when comparing with vanilla BC, the total number of transitions may differ even with the same number of episodes. To address this, they provide an additional baseline, BC-ext, which corrects for this discrepancy.

- The paper includes both real-world experiments and theoretical analysis.

Weaknesses

- The experimental environments are relatively simple. The impact of the proposed blindfolded expert approach would be significantly greater if it could be shown to work on more complex manipulation tasks, where imitation learning is more commonly applied.

- It would be beneficial to provide more qualitative results beyond just the number of steps. For instance, does the blindfolded expert actually explore a meaningfully larger portion of the state space?

- For the theoretical analysis, the authors reformulate the problem as a classification task. As a result, it is unclear how well the derived bound extends to continuous action spaces, which are typical in robotic manipulation.

- While blindfolding promotes generalization by encouraging exploratory behavior, it can also result in sub-optimal expert demonstrations. This increases the expert’s sub-optimality term in the generalization bound, which could potentially weaken the bound overall.

---

> ### Author Rebuttal · Authors · 2025-07-31
>
> We thank the reviewer for the thoughtful comments and for recognizing the key strengths of our work in its novelty, theoretical analysis, and experiments. Below we provide detailed answers to the questions and comments.
>
> **"Experimental environments are relatively simple".**
>
> Our work tackles generalization across tasks, and therefore, *we focus on domains in which generalization is the main challenge*, rather than other factors that may confound the analysis of generalization (such as control or the need to plan for long horizons). In our experiments, most of the complexity stems from the limited variability of training tasks, which produce a cloned policy that memorizes the data. The resulting model lacks the ability to generalize to unseen test tasks - which we show can be resolved with the introduction of blindfolds (Figures 3 and 5).
>
> Nevertheless, note that the peg insertion task is not as easy as it seems: It is a contact-rich manipulation task -- not trivial, and much harder than common pick-and-place manipulation tasks.
>
> Regarding the maze experiment, **we provide an additional experiment of Procgen Heist game**, in which the levels are more challenging compared to a standard maze. At each level, the agent needs to collect a gem hidden behind a network of color-coded locks accessible through keys (of corresponding colors) that are scattered throughout the level (See [1] for more details on the environment setting). This requires multi-stage planning in the form of lock-waypoints and overcoming variation in sub-tasks (key color) ordering. We apply the blindfold similarly to the Procgen maze and follow the same experimental setting as in the paper. In Table 1 below, we compare the results of the agent trained with the fully informed expert's data ($\pi_{BC}$) to the results of the agent trained on the blindfolded experts' data ($\pi_{BF-BC}$). While the training performance is above 90% for both agents, $\pi_{BF-BC}$ significantly outperforms $\pi_{BC}$ on unseen test levels (43% vs. 7%), which indicates better generalization performance (results obtained over $4K$ trajectories from $200$ different seeds).
>
> *Table 1: Performance comparison on the Procgen Heist experiment. The mean and std are computed over 10 seeds. Top performer in bold.*
> | Method | Train | Test |
> |--------|-------|------|
> | $\pi_{BC}$ | 9.41 ± 0.20 | 0.69 ± 0.44 |
> | $\pi_{BF-BC}$ | **9.91 ± 0.04** | **4.27 ± 1.01** |
>
> **"It would be beneficial to provide more qualitative results".**
>
> Thanks for the valuable suggestion. **We computed two additional metrics**: Map coverage score (Table 2) and the entropy of state visitation (Table 3).
>
> *Map coverage score* is the ratio $C=N_v/N_{total}$ given by the number of visited states $N_v$ divided by all accessible states $N_{total}$, averaged over all episodes. For the peg insertion experiment, we consider the rotation of the robotic arm around the Z-axis as the crucial component of the state space for obtaining the correct articulation for insertion. We compute the ratio of the rotation performed (in radians) divided by $2\pi$ in each trajectory, averaged over all trajectories.
>
> *Entropy of the state visitation* is defined by $H= -\sum_s p(s)\log p(s)$. We calculate $p(s)$ using a histogram (with 20 bins) of rotation angles along the trajectory, averaged over all trajectories.
>
> The results confirm that blindfolded experts explore a larger portion of the state space to compensate for the redacted information in the observations.
>
> *Table 2: Map coverage score of the trajectories demonstrated by fully-informed experts (expert) and blindfolded experts (BF-expert).*
>
> | Mode | hexagon | star | square | plus | triangle | Average |
> |------|---------|------|--------|------|----------|---------|
> | expert | 0.078 | 0.113 | 0.150 | 0.153 | 0.191 | 0.137 |
> | BF-expert | **0.114** | **0.259** | **0.267** | **0.270** | **0.327** | **0.247** |
>
> *Table 3: Entropy of the state visitation of the trajectories demonstrated by fully-informed experts (expert) and blindfolded experts (BF-expert).*
>
> | Mode | hexagon | star | square | plus | triangle | Average |
> |------|---------|------|--------|------|----------|---------|
> | Expert | 2.876 | 3.121 | 3.266 | 3.228 | 3.418 | 3.182 |
> | BF-Expert | **3.100** | **3.416** | **3.546** | **3.577** | **3.631** | **3.454** |
>
> **Extension of the theory to continuous action space.**
>
> Robotic manipulation tasks typically allow for modeling with discrete action spaces due to the inherent smoothness of the physical world. Notably, even advanced state-of-the-art models (e.g., OpenVLA [2]) are trained by mapping continuous actions into discrete tokens. This makes our theory applicable to the experimental setting in the paper and many other settings. We do, however, see the value of the reviewer's suggestion to extend the analysis to a continuous action space. That said, the theory of imitation learning with continuous actions is known to be highly non-trivial, and hardness-analysis has been derived recently [3]. We leave settling this open question to future work.
>
> **Q1. Optimal trade-off between generalization and sub-optimal behavior. Principled method or insight for finding this optimal trade-off?**
>
> Any BC experiment requires some engineering (of the data collection setup, the human demonstrator instruction, etc.), and we believe blindfolds will become another factor in this list.
>
> We can provide some insights on how we designed the blindfold in practice. Take the peg insertion case, in which the blindfold is a mask on the image observation. We tweaked the mask to hide most information that still allows the human to solve the task. At that point, removing even more information would not have produced meaningful demonstrations.
>
> Further studies on principled methods for the automatic design of the blindfold are a good direction for future work.
>
>
> **References**
>
> [1] Cobbe, Karl, et al. "Leveraging procedural generation to benchmark reinforcement learning." International conference on machine learning. PMLR, 2020.
>
> [2] Moo Jin Kim et al. OpenVLA: An Open-Source Vision-Language-Action Model. 2024
>
> [3] Thomas T. Zhang et al. Imitation Learning in Continuous Action Spaces: Mitigating Compounding Error without Interaction. 2025

---

> > ### Comment · Reviewer_bJoN · 2025-08-04
> >
> > The authors have resolved my concerns, and I have updated my score to reflect this.
> > I believe this paper is solid overall.

---

> > > ### Author Response · Authors · 2025-08-08
> > > **Thank you**
> > >
> > > Thank you for recognizing our efforts.
> > >
> > > Perhaps you'll be interested in an experiment we conducted following reviewer qfwW's suggestion. In the robotic peg insertion task, we applied random noise to the experts' observations instead of masking the hole - leading to increased state entropy and reduced success rate as more noise is introduced (results table included in the latest comment for reviewer qfwW).
> > >
> > > The results show that even such a general form of blindfold significantly outperforms the conventional approach (which doesn't use any "blindfold"), further strengthening the generality of our approach.

---

### Note · Authors · 2025-08-12

We thank the reviewers and ACs for their dedication in reviewing our work.

As final remarks, we want to summarize the discussion and its outcomes:

At the outset, all the reviewers recognize that the paper investigates a novel direction - that unlike existing Behavioral Cloning (BC) approaches, which focus on the learning algorithm, this work is the first to consider the behavioral patterns of the experts themselves.
The work proposes a new concept of "blindfolding" the experts to induce a more exploratory strategy, which we show generalizes better. The reviewers also appreciate our well-founded empirical and theoretical basis.

Specifically, **reviewer qfwW** highlighted the immediate potential for impact of our work on robotics and other fields. **Reviewers bJoN and LUkx** emphasize the work's novelty and relevance, which they agree is convincingly supported by theoretical and empirical results. **Reviewer 29eA** points to the effectiveness and simplicity of the method that can complement algorithmic advancements to improve generalization.

In favor of the reviewers' request for an additional and more complex experiment, **we include an experiment targeting the complex Procgen's Heist game**: we collected human demonstrations, trained the networks, and show the effectiveness of our approach that outperforms standard BC.

To address the comment by **reviewer 29eA** regarding domain-specific blindfolds and **reviewer qfwW's** interest in the trade-off between promoting generalized experts' behavior and experts' performance, **we include an experiment of robotic peg insertion that uses a more general form of blindfold**: we collected human demonstrations with random noise added to the experts' observations instead of masking the hole. This experiment provides further evidence that even a general form of blindfold, such as added visual noise, may significantly outperform the conventional approach.

We believe that the additional evidence, experiments, and answers provided in the rebuttal and discussion clarify the significance of our work.

We hope that by broadening the exposure of this work, future research will focus on the behavioral patterns of human demonstrations and their effect on the cloned policies.

---

### Decision · Program_Chairs · 2025-09-17

**Decision:**

Accept (poster)

**Comment:**

(a) This paper proposes a data collection paradigm for behavioral cloning by intentionally blindfolding human experts by withholding some task specific information during demonstration. This encourages exploratory behavior to infer the task, leading to improved generalization to unseen tasks. The authors provide a generalization bound whose dominant terms increase with the amount of task info available to the expert and decreases with # of demonstrated tasks. A peg insertion task is used for testing, which is a hard continuous control problem, as well as a procgen game environment.

(b) The idea of blindfolding human experts to encourage exploration is a smart idea and could be generally valuable and interesting with lots of follow up studies. There is a bound to show that generalization error grows with task‑information available to the expert and shrinks as the number of distinct training tasks increases. Map coverage and state‑visitation entropy show blindfolded experts explore a larger fraction of state space, aligning with the paper’s hypothesis. The rebuttal and discussion addressed technical questions and added new experiments.

(c) Despite the added Heist results, the task diversity remains limited (peg insertion & two Procgen families). More axes of generalization (object/spatial/visual, language‑conditioned tasks, compositional objectives) would strengthen the case. The main bound is developed under a finite‑action formulation and the extension to continuous control is discussed qualitatively  but not formalized. Perhaps the biggest limitation is that there are no principled, task‑agnostic procedure for selecting the right bottleneck strength. The approach currently requires domain knowledge and tuning. But it is not fair to expect this to be resolved - the space of tasks is extremely large in the real physical world.

(d) Three reviewers (qfwW, bJoN, LUkx) ultimately argue for accept, with upgrades after rebuttal. One reviewer (29eA) remains borderline accept due to task‑specificity of blindfolding.

(e) Authors added Heist and argued why the chosen domains isolate generalization rather than long‑horizon control. This materially improved the paper - residual concern about breadth remains, but no longer blocks acceptance (raised by bJoN, LUkx, 29eA). Authors added map coverage and state‑entropy; this directly addressed the request and supports the mechanism (bJoN). Authors acknowledged limitations and positioned applicability via discretization practices in modern VLA models.